

# Drought and Salinity Intrusion in the Lower Chao Phraya River: Variability Analysis and Modeling Mitigation Approaches

Saifhon Tomkratoke, Siriwat Kongkulsiri, Pornampai Narenpitak, and Sirod Sirisup

Data-driven Simulation and Systems Research Team, National Electronics and Computer Technology Center, Thailand
Phahonyothin Road, Khlong Nueng, Khlong Luang District, Pathumthani 12120, Thailand.

**Correspondence:** Sirod Sirisup (sirod.sirisup@nectec.or.th)

**Abstract.** Saltwater intrusion in the Lower Chao Phraya River (LCPYR) is a significant national concern for Thailand, requiring a thorough understanding and a development of effective prediction systems for current and future management. This study investigates the key drivers influencing saltwater intrusion in the LCPYR. Cross-wavelet analysis was applied to examine the interactions between tidal forces, drought conditions represented by the standardized discharge index (SDI) and standardized

precipitation index (SPI), and salinity levels. Numerical simulations are used to explore the covariation among these drivers further. The results show that hydrological drought, particularly indicated by the SDI, plays a major role in driving the sub-annual to annual variability of saltwater intrusion, while extreme events are largely governed by sea-level oscillations, underscoring the importance of non-tidal sea levels. Local precipitation, as reflected by the SPI, strongly influences salinity levels, at times weakening the usual correlation between salinity and hydrological drought. The numerical model demonstrates high accuracy

in simulating both hydrodynamic and salinity behaviors, validating the cross-wavelet analysis and offering a reliable approach for modeling salinity in this complex estuarine system. These findings offer essential insights to guide management strategies and the development of prediction tools for the LCPYR and surrounding regions.

     **Keywords:** Saltwater intrusion; Lower Chao Phraya River; Drought; Sea levels; Wavelet analysis

## 1   Introduction

The Lower Chao Phraya River (LCPYR, Fig. 1), a crucial segment of the Great Chao Phraya Basin (CPYB), serves as a vital artery for Bangkok and other major cities in central Thailand, providing essential water resources for various sectors. However, the river has increasingly been impacted by climate variability and growing human demand, leading to significant fluctuations in the freshwater required to sustain life, agriculture, and industry. Over the past decade, water scarcity in the LCPYR has become more frequent, while the demand for water continues to rise. This increasing demand is driven by industrial developments,

large-scale agriculture, urbanization in Bangkok and surrounding cities, and the country's economic growth prospects. The severe droughts between 2019 and 2021 have highlighted the escalating water scarcity in the CPYB. During the dry seasons, particularly in boreal winter, the LCPYR experiences critically low runoff due to reduced rainfall. De-spite the limited water flow, substantial allocations are made for irrigation and raw water production, with additional losses attributed to evaporation and other uses. As a result, the river's flow diminishes to levels insufficient to maintain a river-dominated regime, turning



the LCPYR into a tide-dominated regime instead. This shift allows saltwater from the Gulf of Thailand (GOT) to penetrate upstream, causing severe saltwater intrusion in the lower reaches of the river and affecting Bangkok and nearby provinces.

Saltwater intrusion is a natural feature of meso-tidal estuaries like the LCPYR, where tidal ranges fluctuate between 2 and 4 meters. The variability in saltwater levels is driven by the complex interplay of river dis-charge, tidal forces, and sea-level rise Tran Anh et al. (2018); Mulamba et al. (2019). In the nearby Mekong River Delta, for instance, severe salinity intrusion

events are frequently observed during periods of reduced freshwater inflow or drought, compounded by rising sea levels Loc et al. (2021); Nghia et al. (2024); Vu et al. (2024). Additionally, near-surface wind has been identified as a contributing factor to these dynamics Reyes-Merlo et al. (2013).

In the LCPYR, however, saltwater intrusion can sometimes occur independently of drought conditions, as was observed in 2019 and 2021. These anomalies have added complexity to water management strategies for responsible agencies. Previous

studies have introduced salinity models for the LCPYR Wongsa et al. (2015); Kulmart and Pochai (2020); Othata and Pochai (2021); Changklom et al. (2022). More recently, Chettanawanit et al. (2022) and Pokavanich and Guo (2024) have highlighted key drivers of salinity intrusion in the region. However, the intricate dynamics of saltwater intrusion, particularly in relation to drought, local factors, and mitigation efforts, still require a more thorough analysis and model evaluation, which will be addressed in detail in this study.

For example, wind-driven water level changes have been highlighted as influential in saltwater intrusion in meso-tidal estu-aries Zhang et al. (2019). This aligns with the dynamics of the LCPYR, given its connection to the GOT, a region known for standing wave and seiche oscillation effects Tomkratoke and Sirisup (2020). These oscillations, driven by monsoon winds and storm surges induced by tropical cyclones, may contribute to the variability of salinity levels in the LCPYR. Long-period tidal influences, such as fortnightly salinity intrusion, have also been observed in estuaries worldwide Garcés-Vargas et al. (2020);

Pokavanich and Guo (2024).

Another local factor includes inflows of excess rainfall or floodwaters from adjacent watersheds and retention areas, which can significantly dilute saltwater content in the LCPYR despite a local drought condition. For instance, during the 2020 drought, floodwaters discharged from Bangkok, located at the lower end of the LCPYR, significantly reduced salinity levels in the river. Such events typically occurred during breaks in the monsoon season, when most of Thailand experienced dry conditions.

Conversely, a reduction in freshwater supply during the dry season can exacerbate saltwater intrusion. However, their impacts on the LCPYR remain undocumented.

Addressing saltwater intrusion in the LCPYR and other major rivers is of national importance to Thailand. Effective solutions require a solid scientific foundation; without it, uncertainties about the causes of intrusion and the most appropriate mitigation strategies persist. Poor management could lead to conflicts among stakeholders. Therefore, preventing conflicts and achieving

optimal solutions should be the primary goal. To this end, relevant communities must be equipped with sufficient scientific knowledge, supported by a robust prediction system capable of assisting with both short-term and long-term projections.

Therefore, this study aims to bridge the knowledge gaps concerning the hydrodynamics and salinity dynamics of the LCPYR. We conduct a comprehensive analysis of the key factors influencing salinity and develop high-accuracy numerical models.



These findings are intended to support water resource management in Thailand and assist communities involved in managing
water quality in urban areas.

The rest of this paper is organized as follows: Section 2 discusses the geography and climatic patterns of the study area,
which is the LCPYR. Section 3 provides details of the data used, and Section 4 introduces methods for performing wavelet
analyses of the salinity contributing factors. Section 5 covers the setups of the numerical simulations for the LCPYR. Section
6 offers the results and discussion. Finally, Section 7 presents the conclusions of this study.

## 2  Study Area

The LCPYR has been a vital resource for Thailand since ancient times, supplying freshwater from its northern headwaters to
sustain agriculture, industry, and communities in the lower region. In recent years, water scarcity in the CPYB has intensified,
primarily due to anomalously low rainfall and other contributing factors. This scarcity has led to increased saltwater intrusion,
posing significant challenges for Bangkok and the vicinity. Therefore, a clear understanding of the LCPYR's geological and
environmental context is crucial for addressing this issue.

Our study focuses on the lower portion of the Chao Phraya River (Fig. 1), the largest waterway in the Chao Phraya Delta
and the second-longest river in the region after the 350-kilometer-long Tha Chin River. The LCPYR is one of Thailand's most
distinctive geomorphological features, characterized by its winding course. Originating from the Chao Phraya (CPY) Dam
in Chainat Province, it flows approximately 280 kilometers to the inner GOT. The lower reach, known as the Chao Phraya
estuary, a long, narrow, meandering, and partially mixed estuary Pokavanich and Guo (2024), lies downstream of Phra Nakhon
Si Ayutthaya Province and is roughly 600 meters wide, while the upper reach, above Phra Nakhon Si Ayutthaya Province, is
about 200 meters wide. The riverbed in the lower part lies 5 meters below the mean sea level, whereas the upper part is 5
meters above the mean sea level. The river's major tributary, the Lower Pasak River, extends from the Phra Rama VI Dam in
Phra Nakhon Si Ayutthaya Province, measuring 50 kilometers in length and 75 meters in width, with a riverbed elevation of 6
meters above the mean sea level.

The total inflow to the LCPYR is primarily regulated by the CPY and Rama VI Dams. Observational data show that,
the inflow averages around 80 $m^3/s$ during the dry seasons but can reach up to 2,000 $m^3/s$ during flood periods. However,
metropolitan water extraction reduces this flow by an average of 55 $m^3/s$. Additional lateral flows and other withdrawals also
affect the overall water balance, but limited in situ measurements of these factors introduce uncertainties in assessing the net
inflow.

Rainfall variability in the LCPYR follows the broader pattern of Thailand's mainland climate, characterized by a bimodal
rainfall pattern with two distinct rainy seasons separated by a dry period Tomkratoke and Sirisup (2022). The monsoon break,
typically occurring from mid-June to July, contributes to this variability and can lead to meteorological drought and associated
water scarcity. Low rainfall usually begins in late winter (February) and continues through the pre-monsoon period (March
to May). The wet season in the LCPYR is heavily influenced by large-scale climate patterns, including monsoon troughs and





tropical cyclones. While some regional climate patterns and climate dynamics are well understood, others remain complex and poorly defined.

Geographically, the LCPYR is part of the GOT system, which acts as an effective oscillator for long-wave phenomena such as tidal waves and storm surges. These oscillating long waves significantly influence both the hydrodynamics and salinity levels near the river mouth and even in remote freshwater zones. Salinity in the LCPYR becomes particularly severe during prolonged drought periods, when the influence of long waves is also more pronounced, as will be shown in this paper. This co-variation between salinity, tidal patterns, and storm surges highlights the importance of considering these mechanisms when analyzing the severity of saltwater intrusion. Therefore, to fully understand and accurately model the system, it is essential to analyze the relationships between salinity in the LCPYR, tides, long waves, river flows, and other important factors.

## 3  Data and Methods

The analysis of this study is divided into two parts: time series analysis and numerical modeling. First, the time series analysis focuses on identifying the covariations between salinity and its environmental drivers, including oceanic factors such as tides and extreme sea levels, as well as drought-related conditions like river discharge and rainfall. Second, we develop numerical modeling processes based on the results from the first part. Below, we provide an overview of the key data and processing methods used in the time series analysis.

### 3.1  Water level and sea level data

Water level measurements near the entrance and mid-sections of the LCPYR (Fig. 1), covering periods of intensified saltwater intrusion from 2015 to 2021, are obtained from the National Hydroinformatics Data Center of Thailand (NHC, www.thaiwater.net) and the real-time monitoring system of the Metropolitan Waterworks Authority (MWA, www.rwc.mwa.co.th). Due to significant data gaps and non-physical variations in the datasets from the Chao Phraya River mouth, data from the Tha Chin River mouth, located approximately 30 km away from the LCPYR, is used as a substitute. Since the sea level conditions at the Tha Chin River mouth closely align with those in the GOT, this dataset is a reliable representation. Additionally, for specific statistical analyses, the water level time series are detrended in order to identify long-term trends and remove seasonal or tidal influences.

### 3.2  River discharge data

In this study, daily river discharge measurements taken at 6:00 AM are collected from stations down-stream of the CPY Dam (Q1) and the Rama VI Dam (Q2). The locations of these stations are indicated in Fig. 1. These datasets are provided by the NHC and the Royal Irrigation Department of Thailand (RID, www.rid.go.th). The time series are combined and standardized using the Streamflow Drought Index or SDI Nalbantis and Tsakiris (2009); Arra et al. (2024); Chatklang et al. (2024).



The SDI is calculated based on the following equation:

$$SDI = \frac{Q_D - \overline{Q_D}}{SQ} \quad , \tag{1}$$

where $Q_D$ is the daily average streamflow, $\overline{Q_D}$ represents the rolling mean, and $SQ$ represents the standard deviation of the streamflow. The time series of the SDI for the period from 2015 to mid-2021 is shown in Fig. 2a. Large negative SDI values effectively identify hydrological drought events in the LCPYR, particularly during the severe droughts of 2019–2021 and in

other years.

### 3.3    Rainfall data

Another measure of drought used in this study is the Standardized Precipitation Index (SPI) McKee et al. (1993); Shamshirband et al. (2020); Jahangir and Yarahmadi (2020). Rainfall data are collected from 128 stations across Bangkok, sourced from the Drainage and Sewerage Department's monitoring system (https://dds.bangkok.go.th). The daily rainfall time series is computed

by averaging the data across these stations. The SPI is then calculated based on this area-averaged rainfall using the following equation:

$$SPI = \frac{P_D - \overline{P_D}}{SP} \quad , \tag{2}$$

where $P_D$ is the daily average rainfall. $\overline{Q_D}$ and $SP$ represent the rolling mean and standard deviation of $P_D$, respectively. Fig. 2b displays the time series of the SPI, which is used to evaluate overall meteorological drought conditions across the

LCPYR. Negative values of the SPI less than -1 indicate periods of rainfall deficit, i.e., meteorological droughts, while positive values greater than 1 indicate periods of rainfall excess. The SPI also emphasizes the influence of rainfall-runoff from both the Bangkok area and adjacent watersheds, particularly during heavy rainfall events, which contribute to salinity dilution in the LCPYR.

### 3.4    Salinity data

Near-surface salinity measurements are collected from the Sam Lae station, provided by the Metropolitan Waterworks Author-ity (MWA, www.rwc.mwa.co.th). To analyze the daily covariation between salinity and its drivers, we develop the Extreme Salinity Index (ESI) by extracting the local maxima of the original salinity data. This approach preserves the extrema and emphasizes daily variations in salinity. The climatology of the ESI from 2015 to mid-2021 is shown in Fig. 2c. Qualitatively, extreme salinity events and drought magnitudes, as indicated by the SDI and the SPI, are well correlated, including the extreme

salinity events E1 (July 2015), E2 (May 2016), E3 (March 2017), E4 (December 2019), and E5 (January 2021). However, a more comprehensive understanding of the relationships among these indices can be achieved through a robust analysis method, such as the cross-wavelet transform, which is presented in the following section.



## 4 Continuous Wavelet Transform and Cross-Wavelet Analysis

Understanding the factors controlling salinity in complex estuarine systems like the LCPYR presents significant challenges.
Previous studies have shown that multiple factors, often interacting subtly, influence the dynamic patterns of salinity in es-
tuaries worldwide Wongsa et al. (2015); Kulmart and Pochai (2020); Chettanawanit et al. (2022); Othata and Pochai (2021);
Changklom et al. (2022); Pokavanich and Guo (2024). To better interpret the significance of these factors, it is important to
identify their periodic structures. Classical Fourier analysis, which reveals modes of variability or oscillation in periodic phe-
nomena, is particularly useful in this regard. An extension of the Fourier analysis, which is the wavelet analysis, is frequently
employed due to its ability to ex-tract and visualize the periodic structures of time series data. This method has proven ef-
fective across fields such as hydrology, oceanography, and climatology. Continuous Wavelet Trans-form (CWT) techniques,
commonly used in a wavelet analysis, are particularly advantageous because they offer both time and frequency localizations,
allowing for detections of transient features and oscillations that change over time. For further mathematical details on CWT,
refer to Tomkratoke and Sirisup (2022).

Cross-wavelet analysis provides additional insights when investigating relationships be-tween two time series. Note only
does it measure the strength of covariation between variables but also tracks how these relationships evolve over time and
across different frequency scales. This capability is particularly useful for detecting synchronicity or phase shifts between
variables such as salinity and tidal forces, even when their interactions change over time. Cross-wavelet analysis can also
identify complex, non-linear relationships that may be overlooked by traditional methods, making it well-suited for analyzing
environmental systems influenced by multiple interacting factors.

In this study, we use the WaveletComp package Rösch and Schmidbauer (2016) to conduct both continuous wavelet analysis
on individual time series and cross-wavelet analysis to explore the interactions between salinity and its potential drivers. This
approach allows us to capture multi-time-scale variations, identify the dominant forces influencing salinity dynamics, and
inform the configuration of boundary conditions and parameterization in the subsequent modeling steps.

## 5 Numerical Modeling

### 5.1 Hydrodynamic-mass transport model

To accurately resolve river–ocean interactions and salinity dynamics in the study area, we employ the Semi-implicit Cross-scale
Hydroscience Integrated System Model (SCHISM). SCHISM is an open-source, community-supported model designed for
seamless three-dimensional baroclinic circulation simulations across various scales, ranging from creeks and rivers to estuaries,
shelves, and oceans Zhang et al. (2015, 2016). It utilizes a semi-implicit finite-element/finite-volume method combined with
an Eulerian–Lagrangian algorithm to solve the Navier–Stokes equations in their hydrostatic form. This hydrostatic assumption
simplifies vertical momentum calculations, assuming the pressure at any point is due to the weight of the water above, while
maintaining accuracies of horizontal velocities in large estuarine systems like the LCPYR, where horizontal flows dominate
over vertical accelerations.



SCHISM's hybrid $\sigma$-coordinate transformation, applied in the vertical domain, ensures accurate representation of complex riverbed and bank geometries. Its unstructured horizontal grid system, composed of triangular or quadrilateral elements, provides flexibility in capturing intricate spatial features. The model supports both Cartesian and spherical coordinate systems, adapting to various scales and regions. The numerical algorithm balances higher-order and lower-order methods to optimize stability, accuracy, and computational efficiency, while mass conservation is rigorously enforced via a finite-volume transport

algorithm.

SCHISM's semi-implicit time-stepping approach eliminates the need for mode splitting and relaxes the Courant–Friedrichs–Lewy (CFL) stability constraints, enhancing computational efficiency for faster simulations. Additionally, the Eulerian–Lagrangian Method (ELM) is employed to conserve mass and further relax stability constraints, ensuring accurate resolution of long-wave dynamics and other hydrodynamic processes. Details of the model grids and configurations are provided below.

**5.2 Computational domain and grid design**

To simulate the LCPYR, we designed a computational domain that includes the main canal and the adjacent narrow bank areas, extending from the river mouth upstream to the CPY Dam in Chai Nat Province (BC7) and the Rama VI Dam (BC5), as shown in Fig. 3. Floodplain inundation is not considered in this study; therefore, the floodplain area is omitted from the model geometry. The geometry is discretized into triangular elements of varying sizes, depending on the river's width.

The computational domain covers the entire LCPYR. In the lower part of the domain, we employ triangular elements ranging in sizes from 25 to 50 meters (Fig. 3e). In contrast, finer elements of approximately 10 to 15 meters are used for the remaining regions (Fig. 3c). This approach results in a triangular grid network comprising of approximately 330,000 elements.

Bathymetry data for the domain are obtained from echo sounder depth measurements provided by the Marine Department of Thailand and nautical charts from the Royal Thai Navy. These data are directly incorporated into the computational domain

without being spatially smoothed. The bed topography characteristics of the LCPYR are presented in Fig. 3a, Fig. 3b and Fig. 3d.

**5.3 Boundary conditions**

Fig. 3 illustrates the boundary conditions implemented in the model to capture tidal waves, storm surges, river discharge, and salinity dynamics. Two upstream boundaries are defined at the CPY Dam (BC7) and the Rama VI Dam (BC5), with discharge

data sourced from the NHC. These dams operate using the "water hammer" strategy (Fig. 4a), a method involving the release of water in high-volume pulses to generate hydraulic pressure, helping prevent saltwater intrusion by maintaining sufficient upstream flow.

At the downstream boundary near the Gulf of Thailand (BC1), mean sea levels, tides, and storm surge conditions are applied using processed data from the Tha Chin station. To simulate water abstraction, a constant pumping rate of 55.0 $\text{m}^3$/s is assigned

at Sam Lae station (BC4).

The Bang Keaw station (BC6) plays a crucial role by aggregating water loss from agricultural withdrawals, as shown in Fig. 4c. These withdrawals impact the amount of freshwater available to dilute saltwater intrusion, particularly during dry seasons,





underscoring the importance of managing water abstraction effectively. To reproduce salinity dynamics during the 2019–2021 drought years, side flows and excess rainfall are incorporated as key drivers influencing salinity during dry seasons. Water gates at Singhnat (BC9) and Bang Buatong (BC10) were occasionally opened to release freshwater, as observed during the extreme salinity event in early 2020. Where no data on discharges are available, small testing flows of 0.1–0.2 m³/s are applied. The forcing data collected for this period is presented in the supplementary materials (Fig. S1 in the Supporting Information or SI).

Side flows from the Bangkok Metropolitan Administration (BMA) also plays a critical role in managing salinity. During severe droughts, flows from the Bangkok catchment can reach 100.0–150.0 m³/s, exceeding the combined discharges from the CPY and Rama VI dams. These side flows significantly dilute saltwater in the estuarine portion of the LCPYR. A typical dry-season flow pattern of 50.0 m³/s is used to represent baseline conditions (Fig. 4d), though flows may increase significantly with summer rainfall. Open boundary conditions are set at the Makasan Drainage Station (BC2) and Bang Sue Station (BC3) to account for BMA side flows.

For salinity modeling, time series data from Wat Ban Paeng station are adjusted and applied to the up-stream boundaries (BC5 and BC7). Due to unavailable salinity data for the inner GOT, the GOFS 3.1 reanalysis product (HYCOM) is used for downstream boundary conditions (BC1). In scenarios with side flows, salinity concentrations of 0.2–3.0 g/L are assigned at BC9 and BC10. Additionally, salinity data from the Port station (MWA) is used to adjust the BC2 boundary.

### 5.4 Initial conditions

The initial conditions for hydrodynamics and salinity are set using the cold-start mode of the SCHISM model. The velocity fields $(u, v, w)$ are initialized to zero. For water elevation $(h)$, daily average above the mean sea level (aMSL) water levels from the observation stations (as shown in Fig. 1) are spatially interpolated onto the computational grid as nodal values.

Setting the initial state for salinity is more challenging because an improper starting point could cause divergence in the salinity model's outputs. To ensure stability, the best approach is to begin with well-mixed and mostly uniform salinity conditions, such as those typically observed at the end of the rainy season (usually in October).

### 5.5 Model calibration and validation

In our view, validation studies that focus on short to medium durations (e.g., 7–30 days) may capture daily or sub-daily salinity variability but often overlook longer-term seasonal variations. This limitation can obscure a model's ability to fully predict the range of saltwater intrusion dynamics. For the LCPYR, seasonal variability of saltwater, especially during the dry season, is the most critical factor, as severe intrusion is not a concern during the wet season, which instead promotes the dilution process. Accurately reproducing both short-term and seasonal variations is essential for thoroughly assessing the model performance.

In this study, we focus on a long-term simulation period of six to seven months, encompassing the dry season when saltwater intrusion is most pronounced. This time frame is sufficient for capturing seasonal variability and evaluating the model's performance during the critical stages of saltwater intrusion: development, peak, and recession.

To facilitate the evaluation of model performance and interpretation of errors, we divide the model results into two experiments based on different mitigation strategies:



- **Experiment A:** Water hammer combined with diverting discharges (November 1, 2019, to July 31, 2020).

- **Experiment B:** Water hammer without diverting discharges (November 1, 2020, to May 5, 2021).

These periods cover the crucial dry season when saltwater intrusion is the most significant. Observed water level, river flow, and salinity data are used to validate the model. Stations with minimal missing data and without non-physical fluctuations are selected for validation. The selected stations are shown in Fig. 1; Sam Lae, Wat Sai Ma Nuea, Memorial Bridge, Khlong Lat Pho and South Bangkok stations are equipped with both salinity and water level sensors.

For river flow validation, data from the Rama VIII Bridge station (https://dds.bangkok.go.th), Wat Kema (www.rwc.mwa.co.th), and Bang Sai (www.rid.go.th) stations are used, as indicated in Fig. 1. To assess model accuracy, we calculate the model skill score (Murphy, 1988), root mean square error (RMSE), and Pearson correlation coefficient.

## 6   Results and Discussion

### 6.1   Influence of tidal and non-tidal factors on salinity variability

This section examines the relationship between water levels and salinity in the LCPYR using cross-wavelet analysis. We analyze water levels near the inner GOT at Tha Chin station (T1 in Fig. 1) and salinity levels in the middle LCPYR at Sam Lae station (S5 in Fig. 1) to understand their covariation patterns over time (Fig. 5). The cross-wavelet spectrum reveals significant covariation between salinity and both tidal components, including semi-diurnal and diurnal tides, and non-tidal influences. The periodicity structures can be broadly categorized as below:

**Annual and sub-annual modes:** The annual mode, characterized by a band of high spectral power surrounding the time scale of 356 days (Fig. 5a-b), is a fundamental contributor to salinity variability, persisting throughout the study period. Sub-annual modes, such as the half-year cycle, also play a significant role in intensifying covariation during events like in E1, E4 and E5.

**Fortnightly modes:** Intermediate timescales around 14 days also show significant covariation with salinity extremes in the LCPYR. Notably, the spectral power of these modes is close to that of the tidal modes, particularly during events like E4 and E5. The significance of these modes becomes clearer when analyzing trend-removed water level data (Fig. 5c).

Weekly and sub-weekly modes: Intermittent covariation at shorter time scales (4–7 days) is observed in Fig. 5c, with periods shorter than 4 days displaying more irregular patterns. These shorter-term modes also influence salinity intrusion events, particularly during E4 and E5. The significance of these modes, along with the other spectral components, underscores their role in the development of extreme salinity events in the LCPYR.

**Tidal modes:** The analysis in Fig. 5 also shows high spectral power centered on diurnal and semi-diurnal time scales (approximately 1.0 and 0.5 days, respectively). These active phases indicate a strong covariation between salinity and tides, with a slight phase lag of about 30 minutes, as indicated by the northward arrows. The covariation is most pronounced during spring tides, with distinct active phases occurring during E1, E2, E3, E4 and E5. These periods coincide with severe drought conditions, highlighting the increased influence of tides during low-flow periods. Conversely, during wetter years such as





2017 and 2018, and even most of 2019, the covariation weakens, indicating reduced tidal dependence for saltwater intrusion. When the water levels and salinity at Sam Lae station are analyzed directly (Fig. 5b), the results display a similar periodic

structure at diurnal and semi-diurnal time scales. However, more in-phase relationships emerge, as indicated by the eastward and northeastward arrows, with a shorter lag period of approximately 15 minutes. This observation reinforces the strong influence of tidal forces on salinity levels in the LCPYR.

Cross-wavelet analysis between water levels at Tha Chin station and salinity at Sam Lae station (Fig. 5a) and between water levels and salinity at Sam Lae station (Fig. 5b), which is in the middle of the LCPYR, re-veals the hidden role of oceanic

long waves in driving salinity dynamics in the LCPYR. A combination of both tidal and non-tidal modes is necessary to fully explain the salinity variability. While daily and shorter-period variations are largely driven by tidal effects, longer-term sea level fluctuations including fortnightly, sub-annual, annual modes and others contribute significantly to the development of extreme salinity events. This finding underscores the importance of understanding salinity variability beyond just tidal influences. Further investigation into other key factors, such as drought conditions, is essential to fully comprehend the mechanisms

behind salinity intrusion in the LCPYR. Evidence of these relationships will be discussed in the next section.

## 6.2 Influence of hydro-meteorological drought on salinity Intrusion

This section explores the variability of salinity in relation to hydrological and meteorological drought conditions. We analyze the combined discharge from the upstream CPY and Rama VI Dams, standardized as the Streamflow Drought Index (SDI), alongside salinity levels measured at the water abstraction unit in the middle LCPYR at Sam Lae station, a represented by the

295 Extreme Salinity Index (ESI). Fig. 6 presents the periodic structure of covariation between SDI and ESI, revealing patterns at weekly, monthly, and yearly timescales. These patterns offer insights into the origins and peak periods of saltwater intrusion events. For clarity, the periodic structures are categorized as follows:

**Annual mode:** In Fig. 5d, the high spectral power centered around the annual timescale shows a strong covariation between salinity and hydrological drought conditions. Initially, the phase relationship demonstrates a drought-leading characteristic

(indicated by westward and southwestward arrows). However, after 2019, this shifts to a salinity-leading pattern (northward arrows), likely due to the intensification of drought conditions (SDI below -5.0, Fig. 2a) following the extreme salinity peak in January 2020 (Fig. 2c). Events E4 and E5 appear to be driven by this drought propagation mechanism.

**Sub-annual mode:** High spectral power with a half-year period (182 days) and its surrounding timescales is also prominent, particularly between 2019 and 2021. As shown in Fig. 5d, the drought-leading relationship is more pronounced in this mode

compared to the annual mode. Shorter timescales, around 112 days and below, exhibit intermittent significance. The intensification of spectral power around the half-year mode and shorter periods aligns with the E4 and E5 events. Possibly, these sub-annual periods and shorter timescales likely stem from dam discharge regulations, such as the "water hammer" effect. Despite large negative SDI values during E4 and E5, a strong covariation with ESI is maintained, suggesting that water management plays a significant role in handling saltwater intrusion, although other contributing factors must also be considered.

To further explore the influence of drought on salinity variability, we analyze the Standardized Precipitation Index (SPI) and how it covaries with the ESI (Fig. 5e). The annual mode remains dominant, with intermittent significance in sub-annual modes




such as the half-year and shorter timescales. The phase relationship in the annual mode shows a more pronounced drought-leading pattern (westward arrows), particularly between 2019 and 2021. The weak and active phases of the half-year mode resemble those of the SDI, but the SPI exhibits stronger covariation at shorter timescales (shorter than 112 days), providing additional insight into extreme salinity events like E4. In contrast, the shorter timescales for the SDI exhibit more intermittent behavior, suggesting that meteorological drought indices like the SPI are crucial for understanding the origins of extreme salinity intrusion in the LCPYR.

The lack of correlation between salinity and the Streamflow Drought Index (SDI) in 2020 offers critical insights into the dynamics of saltwater intrusion in the LCPYR. Despite severe drought conditions, indicated by low SDI values in mid-June 2020 (Fig. 2a), salinity levels at Sam Lae station receded instead of increasing as expected. This discrepancy raises questions about whether the low discharge rates (60–65 m³/s) were enough to mitigate intrusion or if additional factors played a role. Possible explanations include reduced agricultural water withdrawal or successful mitigation efforts during that period. Additionally, discharges from Bangkok's catchment areas likely contributed to further salinity dilution downstream, adding complexity to the relation-ship between drought and salinity.

To explore these dynamics, we analyzed rainfall variability during the salinity recession periods of 2020 and 2021 (Fig. S2). Rainfall deviated from climatological norms, especially in 2021, when rainfall over Bangkok exceeded 100 mm between April and May (Fig. S2b). Although the exact causes of these anomalies remain outside the scope of this study, we hypothesize that intensified southerly winds, transporting moisture from the Gulf of Thailand (GOT), may have contributed to the unusual rainfall in central Thailand, as seen in 2019 and 2020. Meteorological drought, represented by the Standardized Precipitation Index (SPI), likely exacerbated water losses, intensifying salinity in the freshwater zone. In contrast, anomalously high rainfall over Bangkok during these periods could explain rapid salinity dilution. Runoff, sometimes exceeding 100 m³/s, was discharged into the LCPYR through drainage systems at Bang Sue and Makkasan (BC3 and BC2, respectively, in Fig. 3a), further diluting salinity downstream. These SPI-driven rainfall patterns provide further insight into the complex salinity dynamics observed in the lower LCPYR.

Our analysis identifies key drivers of salinity intrusion during 2015–2021. Oceanic factors and drought conditions play crucial roles in modulating sub-annual salinity variability. However, the wavelet analysis did not pinpoint a definitive origin for the phenomenon. We hypothesize that hydrological drought acts as a fundamental driver by shifting the river to a tide-dominated regime, which enables non-tidal sea levels to push salt-water into the freshwater zone. Tidal waves then amplify salinity extremes. Increasing upstream inflow dilutes salinity, while urban runoff significantly accelerates the dilution process. These findings will be further validated in the numerical simulations in the following section.

## 6.3 Numerical Simulations and Sensitivity Analysis

In this section, we present the results of our numerical experiments conducted to support our data analysis and validate the interpretations made in earlier sections. Our goal is to deepen our understanding, ultimately facilitating the development a numerical model used for salinity prediction with high accuracy in the LCPYR. We employ the state-of-the-art SCHISM model Zhang et al. (2015, 2016) as the primary tool for this analysis. With the SCHISM model properly configured, we





accurately reconstruct the hydrodynamics and salinity dynamics during the 2019–2020 and 2020–2021 periods. The key results are presented below.

### 6.3.1 Hydrodynamic and salinity model validation

Our initial focus is on modeling the hydrodynamic phenomena in the LCPYR, starting with the validation of the model during

one of the most extreme salinity periods (from late 2019 to early 2020). Once the model's performance is confirmed, we proceed with modeling the salinity dynamics. Validation results for the hydrodynamic properties, including water levels from various stations along the river, are presented in Fig. 6 and Table 1.

Capturing the unique dynamic mechanism of tidal transformation highlights the model's accuracy and validates its configuration. In Fig. 6, the decrement of higher high water (HHW) is clearly visible as it progresses upstream. In contrast, the lower

high water (LHW) shows the opposite behavior, increasing slightly upstream. This shift emphasizes the intricate dynamics between tidal propagation and river flow. The model accurately captures two significant events of extreme salinity, occurring on December 28, 2019, and January 13, 2020, as marked by the blue vertical lines in Fig. 6. These events are critical to understanding the extreme salinity conditions in the LCPYR. These results underscore the robustness of the SCHISM model in simulating tidal dynamics and predicting critical salinity events.

Considering the resonant properties of the GOT Tomkratoke et al. (2015), the HHW is primarily induced by the most energetic tidal components, particularly the diurnal tides (e.g., K1 and O1 in Fig. 6 of Tomkratoke et al. (2015)). In contrast, the LHW is associated with the oscillation of semi-diurnal tidal signals (e.g., S2 and M2 in Fig. 6 of Tomkratoke et al. (2015)), which may be linked to the formation of the fortnightly tide inside the estuary, driven by the modulation of semi-diurnal tides Hill (1994); MacMahan et al. (2014). Our results demonstrate that the transformation mechanisms of tidal waves in the

LCPYR are accurately resolved by the current solver. This indicates that the model's domain design, decomposition, turbulence model configurations, numerical scheme selection, and balance between dispersion and diffusion are all properly determined. Additionally, the open boundary forcing data (ocean-side) are appropriately prescribed. Overall, the model achieves a high correlation coefficient (r) and skill score (SS) of over 0.90, with a low root mean square error (RMSE) of 0.10 meters or less at every tide station considered (Table 1(a)).

For tidal current validation, although limited data are available, a 54-day time series of water flow from April to June 2019 was used to validate the model results. The locations of the flow observation stations are shown in Fig. 1. Hourly flow rates are measured at two stations, Rama VIII Bridge (C1) and Wat Kema (C2), and daily flow rates, measured at 6:00 AM, is from Bang Sai station (C3). Comparisons of the observed and modeled river flows are presented in Fig. 6.

As shown in Table 1(b), although the model achieves good correlation coefficients (r) ranging from 0.86 to 0.89 and model

skill scores (SS) between 0.7 and 0.94, it tends to underestimate the highest tidal current magnitudes. Consequently, the RMSE for the modeled tidal currents reaches 400.0–500.0 m³/s. We speculate that the performance of the tidal current simulation can be improved in future work, particularly by incorporating updated bathymetry data and refining the vertical layers of the model. Despite this limitation, the model successfully captures the overall dynamic behavior of the tidal currents, which is sufficient to





represent the key hydrodynamic mechanisms during the tide-dominated phases in the study area. Given this balance, we accept
the current level of model discrepancy and proceed with the salinity simulation.

Fig. 7 compares the observed and modeled salinity, focusing on variation patterns to interpret the simulation's behavior.
Daily-scale variations correspond to tide-controlled processes, while longer periods are attributed to upstream discharge effects
and non-tidal influences, as discussed in the previous section on wavelet analysis.

Fig. 7a-7e demonstrates the model's accuracy in representing the extreme salinity intrusion from December 2019 to July
2020. The model accurately captures the tidal modes of salinity variability. Although occasional small overestimations or
underestimations occur over the extended simulation period, the model resolves the long-term recession trend in salinity with
high precision (Fig. S3). Table 1(c) shows that the correlation coefficients and model skill scores reach values as high as 0.90
for most stations. The RMSE values at Sam Lae, Wat Sai Ma Nuea, Memorial Bridge, Khlong Lat Pho, and South Bangkok
stations are 0.08, 0.40, 0.86, 1.54, and 3.22 g/L, respectively.

One of the most extreme salinity intrusions in the last decade occurred from November 2020 to June 2021. As shown
in Fig. 7f-7j, the model accurately captures the dynamic behaviors at the daily timescale and demonstrates strong potential
for long-term forecasting. Table 1 indicates high correlation coefficients and model skill scores for the salinity prediction,
exceeding 0.90. However, in the early stages, the model slightly underperforms in capturing the salinity maxima at lower
stations, resulting in a modest increase in RMSE. As the simulation progresses over the longer time span, these discrepancies
diminish. Additionally, the modeled salinity time series' recession trends closely match the observed data across all stations
(Fig. S4).

During June to July, 2020, and April to May, 2021, the recession trends in salinity closely align with the characteristics of
the BMA flood discharges, particularly the contribution from the Bang Sue tunnel (Fig. 4d and Fig. S1d). The rapid increase
in the flow rate is accompanied by a significant reduction in salinity levels (Fig. 4f and Fig. S1f). This observation further
supports the interpretation of the major drivers behind the periods of non-dependence between salinity and drought conditions,
as discussed earlier.

### 6.3.2  Sensitivity analysis of the model

In this section, we investigate the sensitivity of the model's response to variations in boundary condition settings. Specifically,
we focus on how changes in freshwater flux, water elevation, and salinity at the boundaries affect model outputs. The aim
is to assess the extent to which uncertainties or errors in these boundary conditions influence the accuracy of the model's
predictions. By analyzing the model's sensitivity to different boundary condition definitions and input values, we can quantify
their contributions to the overall model uncertainty. Ultimately, this sensitivity study helps identify uncertainties arising from
unknown characteristics or values in the model inputs. To achieve this, we alter the boundary conditions from our best-fit setup.
First, we examine the model's response to variations in freshwater fluxes prescribed at the upstream boundary conditions (Fig.
3), side flow inputs, and water losses.

The freshwater flux magnitudes are defined as a net inflow ($Q_n$), which represents a balance of the diverted water and water
losses (Equation 3). The diverted water includes total inflow ($Q_u$) from upstream boundary flux-es and possibly from tributary



rivers ($T_f$) or watersheds ($BMA$). Meanwhile, the water losses include major abstractions ($M_a$) at Sam Lae station and local abstractions ($L_a$) along the river:

$$Q_n = (Q_u + T_f + BMA) - (M_a + L_a)  \quad , \tag{3}$$

where:

$Q_n$ is the net inflow,

$Q_u$ is the total upstream discharge from the Chao Phraya dam (QCPY) and Pasak (Rama VI dam, QPS) rivers,

$T_f$ represents the flows from the Tha Chin River diverted to the Singhnat (TSN) and Bang Buatong (TBBT) gates,

$BMA$ is the treated water discharge from the Bangkok Metropolitan Administration,

$M_a$ is the major abstraction at Sam Lae station, and

$L_a$ represents the local abstractions.

An event of extremely high salinity intrusion during the winter of 2020 highlights the importance of $Q_n$. During the peak salinity in January 2020, the net inflow ($Q_n$) was around 112.0 m$^3$/s, which would typically promote intensive dilution rather than salinity intrusion. However, the intrusion intensified during this period. To explain this unexpected behavior, we must consider the conditions prior to the peak event (before November 20, 2019). At that time, QCPY was 70.0 m$^3$/s, with no contributions from QPS, TSN or TBBT, and water losses exceeding 10.0 m$^3$/s. As noted earlier, the BMA flow can be disregarded when focusing solely on the freshwater zone. During the intrusion's onset, the net inflow was as low as 6.0 m$^3$/s or less, which was insufficient to pre-vent saline water from the GOT from penetrating into the freshwater zone. Our simulation results confirm that this low inflow was a critical factor that allows the saltwater intrusion to occur.

As the intrusion intensified, mitigation practices such as increasing upstream flow and diverting water from tributary rivers into the LCPYR were implemented. During January 2020, the average discharges from QCPY and QPS were 84.0 m$^3$/s and 7.0 m$^3$/s, respectively, making $Q_u$ approximately 91.0 m$^3$/s. The climatological mean for Ma was about 55.0 m$^3$/s, and $T_f$ was estimated at 39.0 m$^3$/s (from discharges at BC9 and BC10). Average BMA flow was 46.2 m$^3$/s, while our best-fit model suggests an average $L_a$ of 9.4 m$^3$/s. Thus, $Q_n$ during winter 2020 was around 112.0 m$^3$/s. Altering this magnitude led to significant increases in the model's RMSE (Table 2). Reducing upstream discharge, increasing water losses, or excluding side flows exacerbates salinity intrusion in the model (Figs. S5–S6).

Water loss, or local abstraction ($L_a$), plays a critical role in salinity intrusion, though its magnitude and variability cannot be directly measured. During the winter of 2019 to summer 2020, our best-fit model suggests a mean La of 9.4 m$^3$/s, with highest values fluctuating between 30.0 and 48.0 m$^3$/s (Fig. 4c). To explore the model's sensitivity, we perturb the abstraction rate at Bang Kaew (BC6), to a factor of 2, 3, and 4 of the La value used in the best-fit simulation. Table 2 shows the RMSE at Sam Lae under different $L_a$ and hence $Q_n$ values. It shows that higher RMSE is associated with higher $L_a$.




Additionally, we perturb the total upstream discharge at two different locations: reducing QCPY by a factor of 0.8, and completely setting QCPY (at BC7) as well as QPS (at BC5) to zero (the three rightmost columns in Table 2). As expected,
reducing QCPY results in increased model errors, especially at Sam Lae, and excluding QCPY leads to the highest RMSE as $Q_n$ is reduced. Conversely, reducing QPS has a less pronounced effect.

We also perturb Qn by perturbing the treated water discharge from the BMA and the flows from the Tha Chin River ($T_f$, $T_{\text{SN}}$, and $T_{\text{BBT}}$). Table 2 also shows that excluding the diverted flows ($T_f$) at boundary conditions BC9 and BC10 lead to a fivefold increase in the model's RMSE for the freshwater zone at Sam Lae. Omitting flows from Bang Buatong or the Bangkok
Metropolitan Administration ($BMA$) has little effects on salinity trends in the freshwater zone (S1 and S2), but errors increase in the tide-dominated region such as Memorial Bridge (S3, also known as Phra Phutta Yodfa Bridge), Khlong Lat Pho (S4) , and South Bangkok (S5). Notably, reducing $BMA$ flow results in higher errors than reducing $T_{\text{BBT}}$, indicating that $BMA$ flow is a critical factor for modeling salinity in the LCPYR. Lowering $BMA$ flow could promote salinity intrusion in the lower reaches of the river. Additionally, reducing $BMA$ salinity concentration by 75% significantly increases the RMSE at most
stations (2PSU-$BMA$, Table 2) and steepens the recession trends of salinity (Figs. S6c–d).

One crucial input for modeling salinity intrusion is sea level. As discussed earlier, the covariation of sea level and salinity during winter 2020 reveals complex behaviors (Fig. 5). We conduct simulations without storm surge components and with altered mean sea levels (MSL) to test the model's sensitivity. The results (the bottom part of Table 2 and Fig. S6d) show that removing the storm surge component (Tide only) increases the RMSE by 1.5 times, while raising the MSL by 1.0 m
(MSL+1.0m) quadruples the RMSE. Although abrupt changes in MSL (e.g., MSL+0.4m and MSL-0.4m) temporarily affect salinity intrusions, they do not significantly alter the overall mean state of salinity (Fig. S6d). Our findings suggest that sea level variability, rather than its absolute magnitude, is critical in influencing salinity intrusion.

In summary, the model's RMSEs demonstrate high sensitivity to upstream inflow, emphasizing its crucial role in influencing salinity dynamics in the LCPYR. The accuracy of the model is particularly affected by fluctuations in QCPY, while the
broader impact of water loss conditions also demands attention. Flows from $T_{\text{BBT}}$ and $BMA$ play a significant role in the tide-dominated region but have less influence on the freshwater zone. Additionally, non-tidal components at the downstream boundary provide temporary effects on salinity, though they still contribute to overall model accuracy. These findings confirm the robustness of our model configuration, presenting a reliable method to reduce uncertainties and achieve high accuracy. Ultimately, we have developed a robust numerical model that forms the foundation of the LCPYR salinity prediction system.
During the extreme event in early 2020, mitigation measures, including water diversions, were implemented to increase $Q_n$ as previously described. However, the efficiency of these measures has not yet been fully evaluated. Excluding the $T_{\text{BBT}}$ flow reduces $Q_n$ to 94.0 m$^3$/s, which has minimal impacts on the freshwater zone. Reducing $T_{\text{SN}}$ flow results in a similar $Q_n$ but amplifies peak salinity and raises the mean salinity across all stations. A 25% reduction in $Q_{\text{CPY}}$ flow, which lowers $Q_n$ to 28.0 m$^3$/s, result in the most severe impacts, significantly elevating the mean salinity. Most scenarios suggest that an effective
$Q_n$ falls within the range of 105.0–112.0 m$^3$/s. If considering only the upstream contributions, i.e., $Q_{\text{CPY}}$, $Q_{\text{PS|}}$, $Q_{\text{SN}}$, the effective $Q_n$ decreases to around 50.0 m$^3$/s. Ideally, the primary inflow should be around 84.0 m$^3$/s for $Q_{\text{CPY}}$, with additional components such as $Q_{\text{PS}}$ contributing 28.0 m$^3$/s, while $M_a$ remains at 55.0 m$^3$/s and $L_a$ under 18.0 m$^3$/s. For mitigation efforts





extending over 30 days, about 328 million cubic meters (MCM) of freshwater would be required. Given the variability of hydroclimatic conditions, exploring alternative strategies remains essential. A key insight from this study is the importance of controlling freshwater discharges to effectively respond to extreme salinity events in the LCPYR.

### 6.3.3 Model applications and scenario testing

In this section, we present a scenario-based study to explore strategies for efficient freshwater use and assess different assumptions to enhance understanding of salinity intrusion mechanisms in the LCPYR. The scenarios, summarized in Table 3, focus on four main areas: water abstraction location, reservoir regulation, side flow utilization, and the influence of non-tidal sea levels.

The baseline scenario replicates the best-fit model used to simulate the salinity intrusion event during the winter of 2021. A key feature of this baseline is the preservation of the freshwater signal generated by the "Water Hammer" technique, which was applied to mitigate salinity intrusion over a 30-day period (January to February, 2021), using approximately 328 MCM of freshwater. Our findings suggest that alternative strategies, which adjust the timing and volume of upstream freshwater inflows, could increase mitigation efficiency by reducing both peak salinity levels and the overall salinity trend compared to the baseline.

We also explore the potential of relocating the water abstraction point as a future mitigation strategy. The results indicate that changing the abstraction location could either exacerbate or reduce salinity intrusion, de-pending on the new location. Additionally, we find a significant relationship between intensified salinity intrusion and certain non-tidal sea level characteristics, which should be considered in future management strategies. Detailed outputs of these scenarios are discussed below.

**Abstraction Location:** Before examining the results, we first outline the model setup for the water abstraction location scenarios. The water abstraction rate of 55.0 $m^3$/s, initially assigned to BC4 (Fig. 3), is adjusted for scenarios involving both northward and southward shifts. For the northward scenarios, the abstraction point is relocated to BC6 and BC8, while for the southward scenarios, it was moved to BC10 and Rangsit. Additionally, a scenario is de-signed where the water flux at BC7 is completely removed to simulate the effects of northward translation. To further explore the response to southward movement, a new boundary condition is introduced at Rangsit. The results of these scenarios are illustrated in Fig. 8a.

Except for the extreme case (BC7-CPY, dashed red line in Fig. 8a), the northward shift scenarios (BC6 and BC8, purple and blue lines, respectively) show a slight amplification of the salinity time series at Sam Lae. Similarly, changes in the mean salinity at Sam Lae and the other stations along the LCPYR are minimal (Fig. 8e). For the southward translation scenarios (BC10 and Rangsit, green dashed and cyan lines, respectively), both the amplitude and mean salinity levels decrease, but this effect is primarily localized within 50 kilometers of the river mouth, up to Sam Lae (Fig. 8e).

**Reservoir Regulation:** We examine the redistribution of the total freshwater volume used in the baseline scenario, focusing on five primary scenarios: CPY_S1, CPY_S2, CPY_S4, RM6_S2, and RM6_S4. These scenarios are described as follows:

1. CPY_S1, CPY_S2, CPY_S4: These scenarios apply the "Water Hammer" technique to the Chao Phraya or CPY Dam (BC7 in Fig. 3). Over a 30-day period, the total freshwater volume for CPY_S1, CPY_S2, and CPY_S4 is 219, 220, and





207 MCM, respectively. While the total volumes are almost similar, the scenarios differ in terms of amplitude and timing of the flow rates (Fig. S7). In CPY_S1, a high flow of 100.0 m$^3$/s is maintained for 2 days, followed by a low flow of 80.0 m$^3$/s sustained over 10 days. CPY_S2 extends this pattern over a longer period while keeping the same fluctuations. In contrast, CPY_S4 features a high flow of 110.0 m$^3$/s for 15 days and a low flow of 60.0 m$^3$/s.

2. RM_S2, RM_S4: These scenarios are applied to the Rama VI Dam (BC5) and in-volve a total water volume of 118 MCM. In RM_S2, a high flow of 70.0 m$^3$/s is main-tained for 13 days, followed by a low flow of 20.0 m$^3$/s, resulting in a total water vol-ume of 123 MCM. RM_S4, representing a scenario with even more constrained freshwater availability, features a high flow of 20.0 m$^3$/s and a low flow of 5.0 m$^3$/s, with a total volume of just 30 MCM.

The results of these scenarios are illustrated in Fig. 8b. Both CPY_S4 (cyan line) and RM6_S2 (green dashed line) demon-

strate a degree of success in reducing peak salinity and its overall trend at Sam Lae. The idealized "Water Hammer" approach, with high flows of 70.0 and 110.0 m$^3$/s, proves effective in limiting salinity intrusion, particularly during periods of extreme salinity. These high-flow phases, occurring about 7 days before and after the peak salinity events (Fig. S7), support the use of a lower flow rate of 20.0 and 60.0 m$^3$/s, for the Rama VI and CPY Dams, respectively. However, the impact remains confined primarily to the tide-dominated region downstream from Sam Lae (Fig. 8f). Scenarios with lower high flows, such as CPY_S1

and CPY_S2 (purple and blue lines, respectively), show minimal improvement over the baseline. In contrast, significantly reduced flows from the Rama VI Dam (RM6_S3) result in amplified salinity peaks and trends, suggesting that maintaining a sufficient low flow is crucial for the dilution process.

Additional experiments are conducted to determine the most effective mitigation practices for reducing the salinity trend from the baseline and to establish the critical flow rates for each river branch. These experiments included modified scenarios

such as Comb_S1, Comb_S2, and Comb_S3, as well as combinations of CPY_S4 and RM6_S2. Other cases with constant flow rates from the Chao Phraya and Rama VI dams (Cnst_S1) and a revised version of the baseline (rBaseline) are also evaluated.

The percent difference from the baseline across all mitigation scenarios is summarized in Tables S1 and S2. The results highlight the importance of the magnitude and duration of total discharge ($Q_u$) during the first 15 days. A significant reduction in the salinity trend can be achieved with the appropriate high flow, potentially allowing for a smaller flow in the remaining

15 days. The regression model for $Q_u$, total water volume, and the mitigation objective is shown in Fig. S8, illustrating a logarithmic relationship. This suggests that simply increasing upstream discharge or freshwater volume does not necessarily enhance mitigation efficiency; instead, these factors must be carefully and precisely managed.

**Application of Side Flow**: The model sensitivity tests highlighted the crucial role of side flows, particularly the $BMA$ discharge, in managing salinity intrusion within the tide-dominated region. Additionally, we evaluate the impact of mitigation

measures, such as releasing freshwater from the Tha Chin River via the Bang Buatong Gate (BC10). However, the dilution effect of this measure appears to be limited, likely due to contamination from saltier water in nearby channels. A more effective strategy would involve controlling the salinity concentration of the diverted water. To assess this, we simulate a scenario with a constant $BMA$ flow of 100.0 m$^3$/s (BMA-100) and a water diversion scenario at BC10 with a constant flow of 10.0 m$^3$/s and a salinity concentration of 0.2 g/L.



As shown in Fig. 8c, the results at Sam Lae indicate that excluding the $BMA$ flow yields similar results to the baseline scenario. However, in the scenario with a 100.0 m$^3$/s flow, the receding trend of salinity in the tide-dominated region becomes steeper (Fig. 8g), reinforcing the significance of BMA flow. Moreover, increasing the flow rate further accelerates the receding trend. The Bang Buatong flow also exhibits a notable effect but is confined to the tide-dominated region. Physically, the Tha Chin discharge rarely exceeds 0.5 g/L, even during extreme saltwater intrusion events such as those observed in early 2020 and
2021. Nevertheless, saltwater contamination from adjacent channels remains a potential issue.

Non-tidal Sea Level Effects: The temporary influence of elevated mean sea levels on salinity intrusion during early 2020 is demonstrated earlier. To further investigate this factor under varying upstream freshwater conditions, we conduct a scenario where the mean sea level is raised by 0.4 meters (MSL+0.4m). Additionally, idealized extreme sea level scenarios, namely Tr 3 days, Tr 7 days, and Tr 15 days, are tested, combining harmonic tides with synthetic storm surges. These storm surges (Fig.
S7c) have amplitudes of approximately 0.5 meters and vary in durations, i.e., 3, 7, and 15 days.

The results of these scenarios are shown in Fig. 8d. As anticipated, a significant correlation between high storm surges and peak salinity levels is observed, particularly in the Tr 7 days scenario. While the effects are temporary, scenarios with longer fluctuation periods, i.e., Tr 7 days, Tr 15 days, and MSL+0.4m, demonstrate stronger suppression of salinity levels. However, as shown in Fig. fig:08h, raising the mean sea level does not cause a sustained increase in salinity beyond 50 kilometers from
the river mouth to Sam Lae.

Saltwater intrusion in estuaries, where tidal forces dominate over river flow, is a widely recognized phenomenon across the globe. In the context of the LCPYR in Thailand, the origins and intensification of saltwater intrusion are particularly complex. The primary driver is hydrological drought, where receding river flows, compounded by anomalously dry conditions often linked to El Niño, play a significant role. Insufficient runoff, combined with uncontrolled water losses, further aggravates the
situation.

Similar to other estuaries, tidal variations and extreme sea levels critically influence the severity of salt-water intrusion. However, the LCPYR also exhibits unique dynamics. Long waves, storm surges, and tidal transformations interact with regional hydrology, amplifying salinity during low-flow periods. Urban discharge, particularly from Bangkok, adds another layer of complexity, as it can either dilute or intensify salinity depend-ing on flow conditions. From these insights, several recommen-
dations emerge for sustainable water management:

1. **Optimizing Upstream Inflow Management**: Adjusting the timing and magnitude of upstream discharges is essential for effective salinity mitiga-tion. Sensitivity analysis demonstrates that optimized inflow management can reduce the volume of freshwater required for dilution by over 30%. Properly sequencing high and low discharge periods ensures that freshwater is used efficiently. Additionally, "swapping" reservoir releases e.g., releasing water from alternative sources
when one dam faces shortages can mitigate the impacts of extreme sa-linity events and serve as a global best practice for water management in estuaries.

2. **Relocating Abstraction Points for Improved Efficiency**: Numerical modeling indicates that relocating water abstrac-tion points away from critical freshwater zones does not necessarily increase salinity intrusion. Therefore, relocating





abstraction units offers a practical strategy for salinity management. This flexibility allows water managers to strategi-
cally withdraw water without compromising downstream water quality or intensifying intrusion in fresh-water zones.

3. **Implementing Water Diversion Strategies**: Diverting water from tributaries, as demonstrated through our model, is
an effective strategy to reduce salinity. The timing, volume, and location of diversions should be optimized to achieve
maxi-mum effectiveness. In particular, urban runoff from the Bangkok Metropolitan Administration has proven to be
an important factor in reducing salinity levels. Ensuring that the salinity concentration of this discharge is carefully
regulated could further enhance its mitigating effect during extreme salinity events. Water diversion strategies offer a
proactive solution to salinity management, especially during severe drought periods.

4. **Developing a Robust Prediction System Using Physics-Based Models**: Physics-based numerical models are essential
for understanding the complex interactions between tides, river flows, and salinity levels. These models also enable the
development of reliable prediction systems, crucial for both long-term planning and real-time management. Accurate
forecasts of sea level variability, particularly storm surges and wind-induced sea levels, are critical to anticipating salinity
intrusion events. Building on prior research, such as studies by Sirisup et al. (2016) and Tomkratoke and Sirisup (2020),
a comprehensive ocean modeling system tailored to the region is recommended to support water resource management.

5. **Assessing and Preparing for Future Climate Scenarios**: Climate change is expected to impact salinity dynamics in the
LCPYR through more frequent extreme weather events, including both heavy rainfall and prolonged dry spells. State-of-
the-art climate models suggest that global warming will intensify hydrological variability, with longer dry periods and
more intense rainfall episodes Narenpitak et al. (2024). These changing patterns, along with rising sea levels and shifting
wind regimes, could significantly affect the extent and severity of salinity intrusion in the LCPYR. Future research should
focus on how these changes will alter the region's hydrodynamics and explore adaptive strategies to mitigate potential
impacts.

A comprehensive approach, combining upstream inflow management, strategic water diversions, flexible abstraction poli-
cies, and advanced numerical modeling, is essential for effective salinity management. As cli-mate variability increases and
water demands grow, these strategies will ensure sustainable water resource management in the LCPYR and provide a frame-
work for addressing similar challenges in other estuarine systems worldwide.

## 7 Conclusions

Saltwater intrusion in the Lower Chao Phraya River (LCPYR) is a major challenge for water resource and environmental man-
agement in Thailand. A comprehensive understanding of the factors driving this phenomenon, coupled with the development
of effective prediction and decision-making tools, is essential for successful management, both now and in the future. This
study examines the primary influences on saltwater intrusion in the LCPYR, employing cross-wavelet analysis to assess tides,
hydrological droughts (via the Standardized Dis-charge Index, SDI), meteorological droughts (via the Standardized Precipita-



tion Index, SPI), and salinity data. Numerical simulations are then conducted to validate the findings and provide a framework for future prediction systems. The key results are as follows:

**Salinity variability modes**: On a sub-annual to annual timescales, salinity variations in the LCPYR are closely tied to sea-level fluctuations, and hydrological and meteorological drought conditions. For the shorter timescales, extreme salinity events are triggered by anomalous sea levels and intensified tidal forces. These findings highlight the crucial role of winter monsoon

winds and the resonant properties of the Gulf of Thailand in producing prolonged seiche events in the LCPYR. However, diurnal, semi-diurnal, and fortnightly tides remain fundamental to overall salinity fluctuations. Additionally, meteorological droughts significantly contribute to the variability of salinity in the LCPYR.

**Numerical Model Performance**: The numerical model based on SCHISM and developed further in this study is proven to be highly accurate in simulating both the hydrodynamics and salinity dynamics. It effectively captures the influence of hydro-

climatic and oceanic factors on salinity patterns, validating the findings from the cross-wavelet analysis. The model suggests that hydrological drought conditions play a more fundamental role as a primary driver for the origination of salinity intrusion. In the end, the importance of other drivers can be proved by numerical experiments, i.e., the sensitivity studies. This model offers a robust framework for understanding salinity dynamics in the region, identifying key mechanisms behind saltwater intrusion, and regulating the dilution process.

**Proposed Mitigation Strategies**: Based on the successful combination of data analysis and modeling, several strategies for mitigating saltwater intrusion in the LCPYR have been proposed. Optimizing the redistribution of freshwater for dilution has emerged as a highly efficient approach for reducing salinity intrusion. The insights gained from this study can support future prediction systems and guide management strategies for saltwater intrusion in the LCPYR, with potential applications for other estuaries in Thailand and neighboring regions.

These findings underscore the importance of combining robust data analysis with advanced numerical modeling to better understand and address saltwater intrusion. As the LCPYR faces increasing challenges due to climate variability and evolving water needs, the tools and strategies developed in this study offer a valuable foundation for proactive management. By optimizing freshwater use and improving the accuracy of salinity pre-dictions, we can mitigate the impacts of saltwater intrusion and enhance water resource management in Thailand and neighboring regions.





**Table 1.** Correlation coefficients (r), model skill scores (SS), and the root mean square error (RMSE) of the modeled (a) water levels, (b) flow rates, and (c-d) salinity, compared with the observational data at various stations.

| Variables | Stations | r | SS | RSME |
|---|---|---|---|---|
| **(a) Water levels** (m) (Dec 2019 - Jan 2020) | **Tide stations** | | | |
| | Sam Lae (S5) | 0.99 | 0.98 | 0.10 |
| | Wat Sai Ma Nuea (S4) | 0.99 | 0.98 | 0.09 |
| | Khlong Lat Pho (S2) | 0.98 | 0.99 | 0.10 |
| **(b) Flow rates** (m$^3$/s) (Dec 2019 - Jan 2020) | **Flow stations** | | | |
| | Rama VIII (C1) | 0.89 | 0.94 | 501.52 |
| | Wat Kema (C2) | 0.91 | 0.95 | 454.76 |
| | Bang Sai (C3) | 0.72 | 0.80 | 288.66 |
| **(c) Salinity** (g/L) (Nov 2019 - Jul 2020) | **Salinity stations** | | | |
| | Sam Lae (S1) | 0.90 | 0.95 | 0.08 |
| | Wat Sai Ma Nuea (S2) | 0.96 | 0.97 | 0.40 |
| | Memorial Bridge (S3) | 0.95 | 0.97 | 0.86 |
| | Khlong Lat Pho (S4) | 0.94 | 0.96 | 1.54 |
| | South Bangkok (S5) | 0.87 | 0.93 | 3.22 |
| **(d) Salinity** (g/L) (Nov 2020 - Jul 2021) | **Salinity stations** | | | |
| | Sam Lae (S1) | 0.91 | 0.93 | 0.13 |
| | Wat Sai Ma Nuea (S2) | 0.99 | 0.99 | 0.39 |
| | Memorial Bridge (S3) | 0.96 | 0.98 | 0.95 |
| | Khlong Lat Pho (S4) | 0.94 | 0.96 | 1.83 |
| | South Bangkok (S5) | 0.87 | 0.91 | 3.75 |



**Table 2.** Salinity root mean square errors (RMSE) in g/L at five different stations for sensitivity simulations representing the extremely high salinity event in early 2020. The best-fit model is also shown for reference. For the sensitivity simulations, different inflow conditions ($Q_n$) are achieved by perturbing various variables on the right hand side of Equation 3: perturbing the local abstraction ($L_a$), perturbing the upstream discharge at the Chao Phraya (CPY) and Pasak (PS) Dams ($Q_{CPY}$ and $Q_{PS}$, respectively), perturbing the treated water discharge from the Bangkok Metropolitan Administration ($BMA$), and removing flows from the Tha Chin River diverted to the Singhnat ($T_{SN}$) and Bang Buatong ($T_{BBT}$), and both ($T_f$). Additionally, various sea level conditions are also perturbed for the sensitivity test.

| | | Salinity RMSE (g/L) | | | | |
|---|---|---|---|---|---|---|
| **Sensitivity simulations** | | Sam Lae | Wat Sai Ma Nuea | Memorial Bridge | Khlong Lat Pho | South Bangkok |
| | $Q_n$ (m³/s) | | | | | |
| **Best-fit model** | 112.0 | 0.13 | 0.85 | 1.02 | 1.43 | 5.29 |
| **Perturbations** | | | | | | |
| $2L_a$ | 100 | 0.18 | 0.64 | 1.11 | 1.80 | 5.99 |
| $3L_a$ | 90 | 0.42 | 0.48 | 1.76 | 3.10 | 6.94 |
| $4L_a$ | 82 | 0.81 | 0.65 | 2.81 | 4.77 | 7.96 |
| $0.8Q_{CPY}$ | 97 | 0.34 | 0.56 | 1.43 | 2.37 | 6.41 |
| No-$Q_{CPY}$ | 28 | 2.94 | 3.49 | 10.13 | 13.50 | 11.89 |
| No-$Q_{PS}$ | 102 | 0.17 | 0.65 | 1.06 | 1.70 | 5.90 |
| No-$BMA$ | 66 | 0.12 | 0.96 | 3.34 | 6.98 | 9.48 |
| 2PSU-$BMA$ | 66 | 0.13 | 1.37 | 2.32 | 2.10 | 5.37 |
| No-$T_f$ | 73 | 0.63 | 1.05 | 4.15 | 6.73 | 8.99 |
| No-$T_{SN}$ | 91 | 0.53 | 0.51 | 1.96 | 3.36 | 7.02 |
| No-$T_{BBT}$ | 94 | 0.13 | 0.57 | 1.78 | 3.25 | 7.02 |
| **Sea level conditions** | | | | | | |
| Tide only | | 0.20 | 1.13 | 1.50 | 1.90 | 5.79 |
| MSL+1.0m | | 0.50 | 1.14 | 2.07 | 3.29 | 7.03 |
| MSL+0.4m | | 0.19 | 0.59 | 1.07 | 1.82 | 6.11 |
| MSL-0.4m | | 0.13 | 0.57 | 1.78 | 3.25 | 7.02 |





**Table 3.** An overview of scenarios for model application and testing, focusing on four approaches: (1) water abstraction locations, (2) reservoir regulation plans, (3) side flow management strategies, and (4) the influences of non-tidal sea levels on salinity intrusion in the LCPYR. The first three approaches focus on assessing changes in the flow rates ($Q_n$) at Sam Lae by perturbing different forcings, while the fourth approach focuses on changes in the water level (m above mean sea level) at Sam Lae when non-tidal sea levels are perturbed. Each of the initial forcings are perturbed at various locations, from BC1 through BC10 and at Rangsit, as shown in Fig. 3.

| Approaches | Scenarios | Locations of perturbatioins | Periods | Flow rates ($Q_n$) |
|---|---|---|---|---|
| | Baseline | BC4 | Water Hammer, as observed | 55 m³/s |
| | BC7-CPY | BC7 | as baseline | 55 m³/s loss |
| 1. Abstraction location | BC6-Bangkaew | BC6 | constant | 55 m³/s |
| | BC8-Noi River | BC8 | constant | 55 m³/s |
| | BC10-Bang Buatong | BC10 | constant | 55 m³/s |
| | Rangsit | Rangsit | constant | 55 m³/s |
| 2. Reservoir regulation | CPY_S1, CPY_S2, CPY_S4 | BC7 | 7-15 days | 60-125 m³/s loss |
| | RM6_S2, CPY_S4 | BC5 | 7-15 days | 5-70 m³/s |
| 3. Side flow | B. Buatong Flow | BC10 | constant | 10-30 m³/s loss |
| | MBA-100, No MBA flow | BC2 | as baseline | +25-50% |
| **Approaches** | **Scenarios** | **Locations of perturbatioins** | **Periods** | **Water levels** |
| 4. Non-Tidal Sea Level | | | | |
| - Mean sea level | MSL+0.4m | BC1 | as baseline | + 0.2-0.5 m |
| - Storm surge | Tr 3 days, Tr 7 days, Tr 15 days | BC1 | 3-15 days | +25-50% |



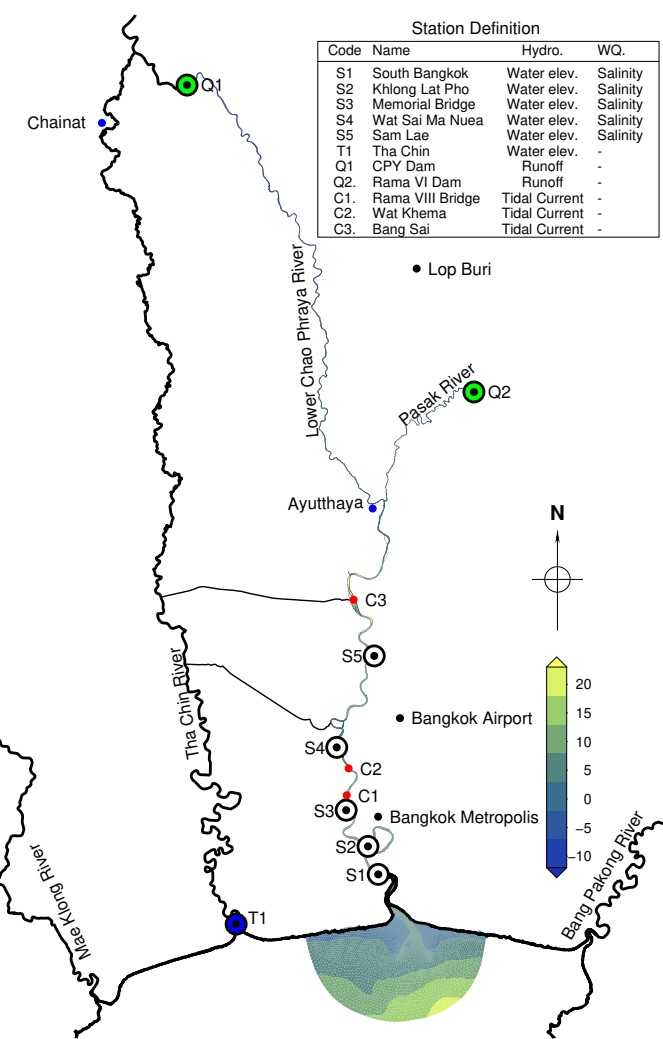

**Figure 1.** The study area, covering the Lower Chao Phraya River (LCPYR), adjacent major rivers, and the inner Gulf of Thailand (GOT). The overlaid bullseye markers represent observation stations. Salinity and water level data are available from the Khlong Lat Pho (S2), Wat Saima Nuea (S4) and Sam Lae (S5) stations. Black dots indicate rainfall stations monitored by the Thai Meteorological Department.





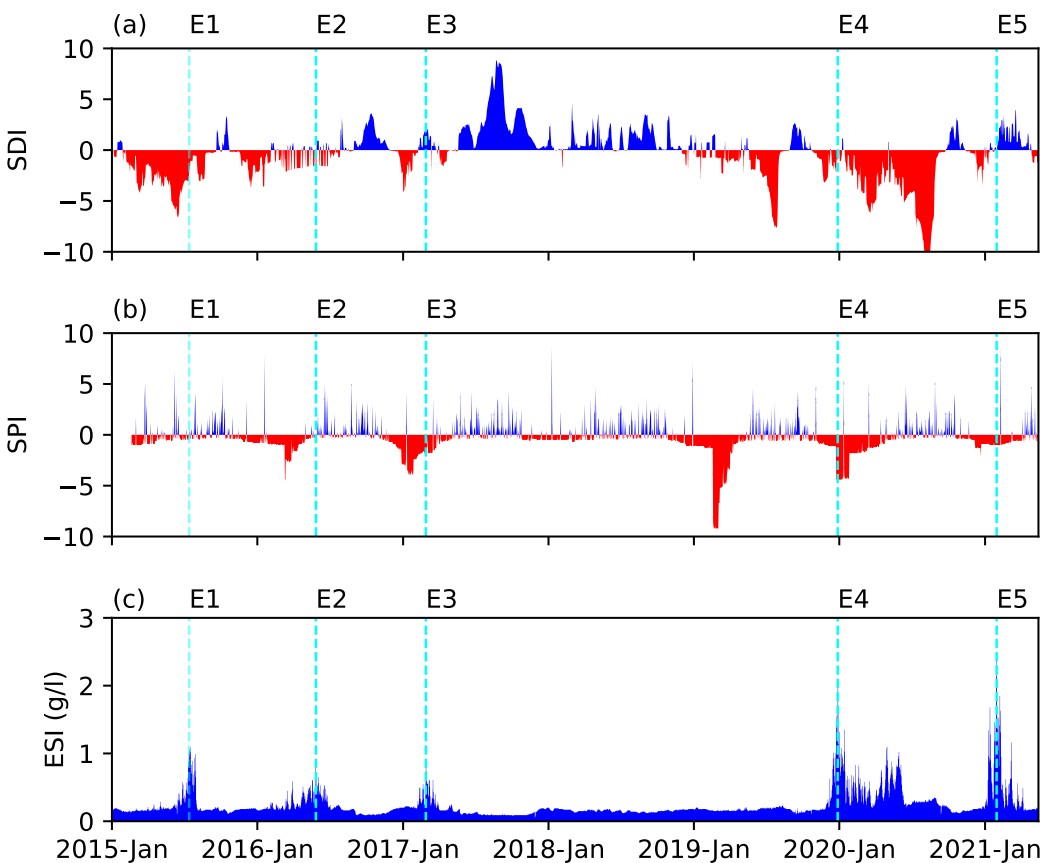

**Figure 2.** Time series of (a) the Standardized Discharge Index (SDI), (b) the Standardized Precipitation Index (SPI), and (c) the Extreme Salinity Index (ESI) from January 2015 to May 2021. The vertical dashed lines and labels (E1-E5) mark the events of extreme salinity during July 2015, May 2016, March 2017, December 2019 to January 2020, and January 2021 to early 2021, respectively.





| Definition of Boundary Conditions | | | |
|---|---|---|---|
| BC1. | GOT | Water elev. | Salinity |
| BC2. | Makasan | Inflow | Salinity (est.) |
| BC3. | Bang Sue | Inflow | Salinity (est.) |
| BC4. | Sam Lae | Abstraction | - |
| BC5. | Rama VI dam | Inflow | Salinity |
| BC6. | Bang Kaew | Abstraction | - |
| BC7. | CPY Dam | Inflow | Salinity |
| BC8. | Noi River | Abstraction | - |
| BC9. | Singhnat gate | Inflow | Salinity |
| BC10. | Bang Buatong gate | Inflow | Salinity (est.) |

**Figure 3.** Computational domain, bathymetry, and boundary condition positions (a) of the Lower Chap Phraya River and zoomed-in views highlighting the bathymetry and domain decompositions of (b-c) the Singhnat Gate (BC9) and the surroundings, and (d-e) the Makasan Drainage Station (BC2) and the vicinity.



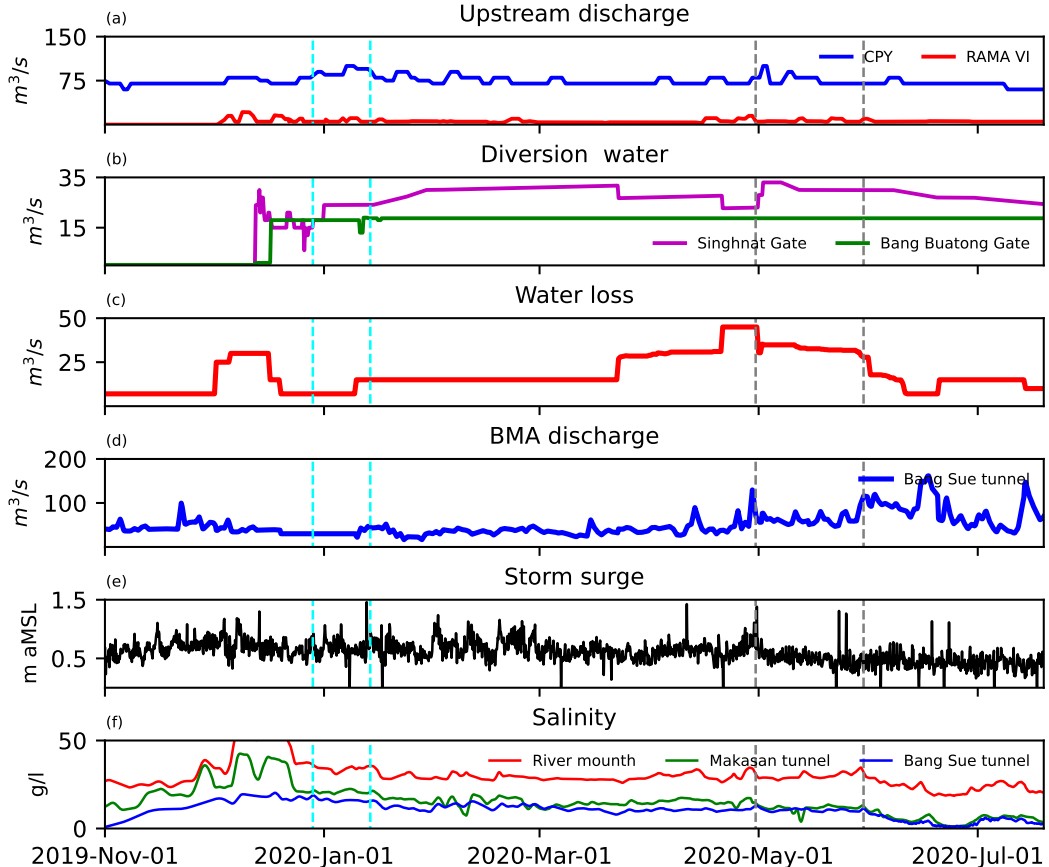

**Figure 4.** Collected forcing data for modeling hydrodynamics and salinity in the LCPYR: (a) the upstream discharge (m$^3$/s) from the Chap Phraya (CPY) Dam (BC7) and Rama VI (BC5) Dam; (b) the diverting discharge water (m$^3$/s) from the Singhnat (BC9) and Bang Buatong (BC10) gates; (c) the water loss (m$^3$/s) at Bang Kaew station (BC6) due primarily to agricultural withdrawals; (d) the Bangkok Metropolitan Administration (BMA) discharge (m$^3$/s) at the Bang Sue tunnel (BC3); and (e) the storm surge (m above average mean sea level or aMSL). The data cover the period from November 1, 2019, to July 2020. Vertical dashed cyan and black lines indicate the extreme salinity events during the winter and summer seasons, respectively.







**Figure 5.** Cross-wavelet power spectrum of paired time series displaying covariation patterns of salinity and its drivers across multiple scales: (a) Water levels at Tha Chin station (T1) and salinity at Sam Lae station (S5). (b) Water levels and salinity observed at Sam Lae station. (c) Same as (b), with trend removed and zoomed into a 64-day period. (d) Standardized Discharge Index (SDI) and Extreme Salinity Index (ESI) spectrum from 2015 to 2021. (e) Standardized Precipitation Index (SPI) and ESI spectrum from 2015 to 2021. The horizontal black dashed lines mark periods of 14, 182, and 365 days. Vertical dashed lines and labels (E1–E5) correspond to key extreme salinity events, as indicated in Fig. 2.



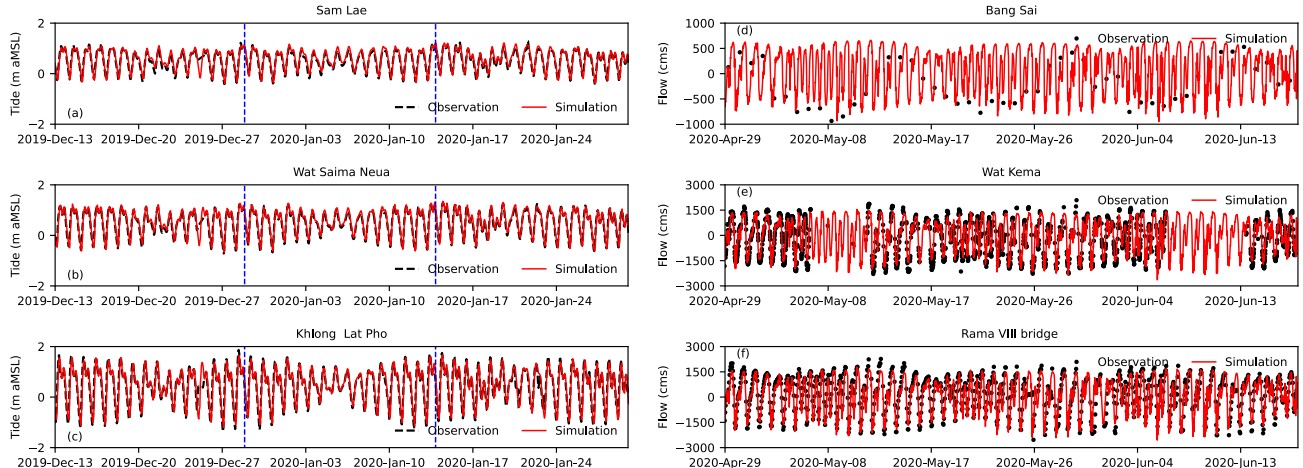

**Figure 6.** Comparisons of observed and modeled water levels and flow rates at key stations along the Lower Chao Phraya River. (a) Observed and modeled water levels at Sam Lae (S5). (b) Wat Saima Nuea (S4). (c) Khlong Lat Pho (S2). The vertical dashed blue lines mark the periods of peak salinity on December 28, 2019, and January 13, 2020. Water levels are shown in meters above mean sea level (aMSL). (d) Observed and modeled flow rates at Bang Sai (C3). (e) Wat Kema (C2). (f) Rama VIII Bridge (C1). Flow rates are presented in cubic meters per second (cms), with black dots represent-ing observations and red lines indicating model outputs.





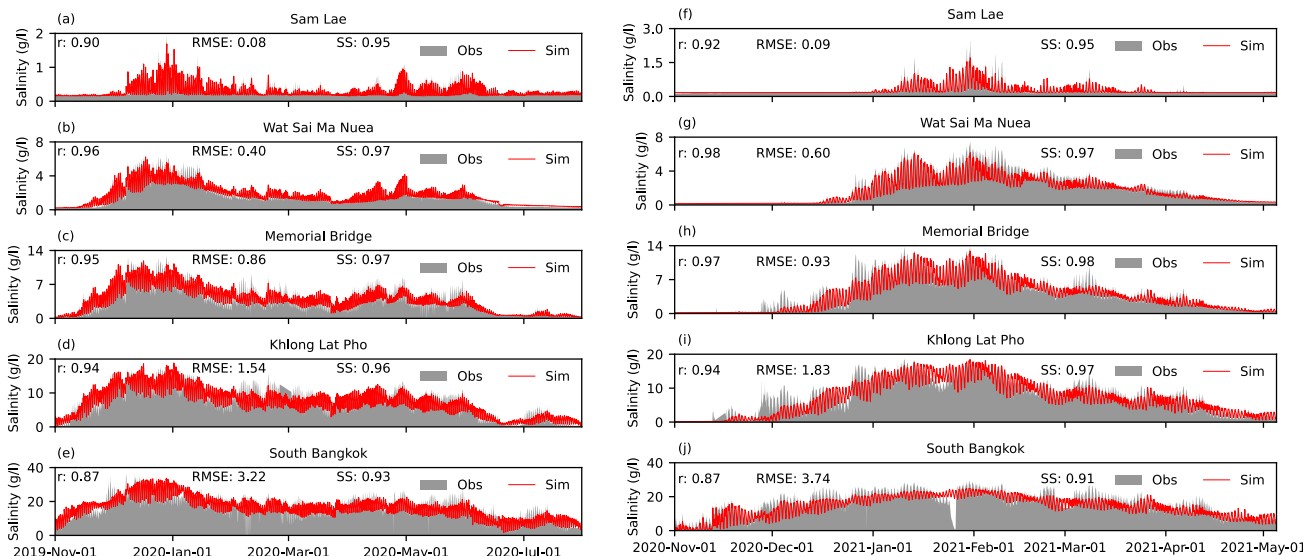

**Figure 7.** Comparisons of observed and modeled salinity levels across key stations for two periods. (a) Sam Lae (S5) for November 2019 to July 2020. (b) Wat Sai Ma Nuea (S4) for the same period. (c) Memorial Bridge (S3) for November 2019 to July 2020. (d) Khlong Lat Pho station during November 2019 to July 2020. (e) South Bangkok (S1) during November 2019 to July 2020. (f-j) Same as in panels (a–e) but showing December 2020 to May 2021 for all stations. Filled grey areas represent observed salinity in g/L, and red lines indicate modeled salinity. Note: Y-axis ranges differ between panels. Station locations are shown in Fig. 1





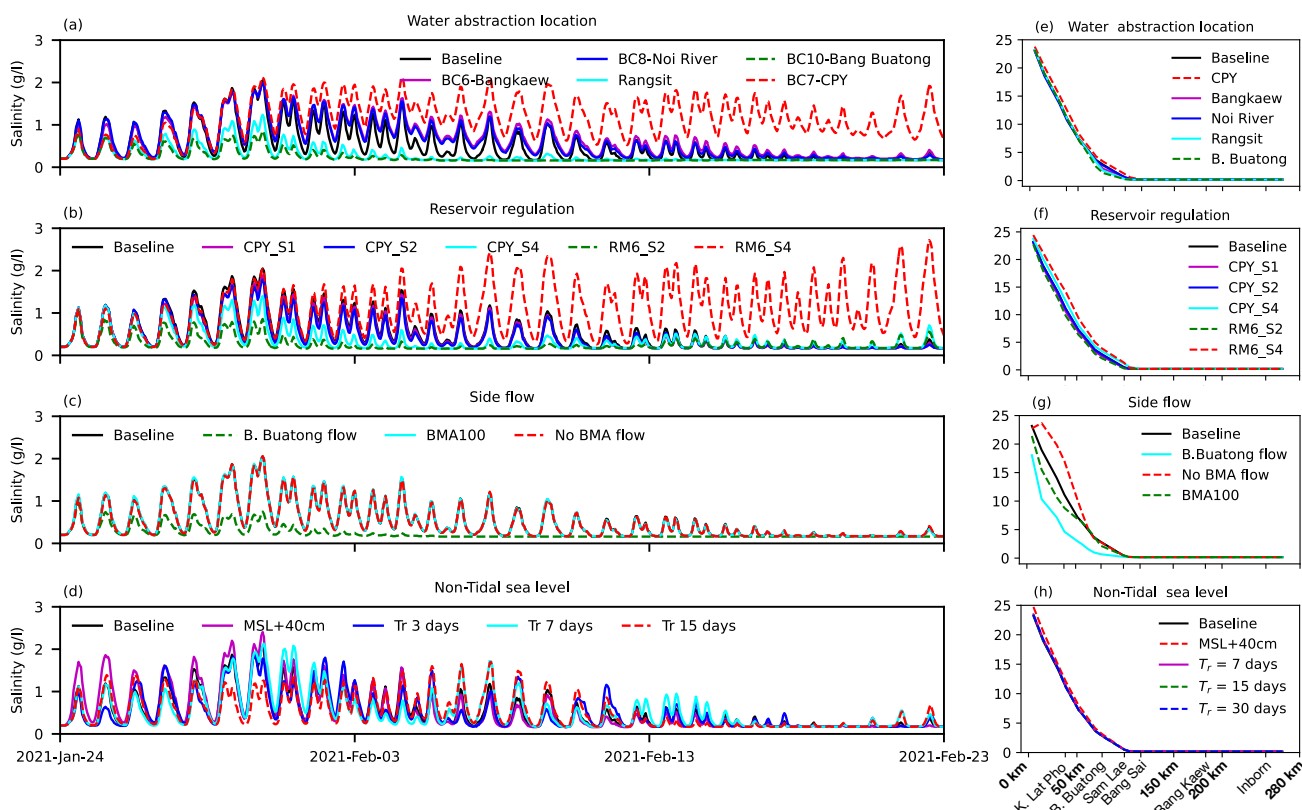

**Figure 8.** Salinity responses at Sam Lae station to different perturbation approaches and their spatial impact along the Lower Chao Phraya River (LCPYR). The left panels (a-d) show results for varying (a) water abstraction locations, (b) reservoir regulations, (c) side flows and diversion water, and (d) non-tidal sea level conditions, based on an extreme event in early 2021. The right panel presents the 24-hour filtered salinity (g/L) along the LCPYR channel, highlighting the impact of sensitivity scenarios listed in Table 3. The distance runs from the river mouth (0 km) to Sam Lae (approximately 280 km).



*Data availability.*  The simulation outputs generated or analyzed during this study are archived at https://nstda-my.sharepoint.com/:f:/g/ personal/sirod_sirisup_nectec_or_th/Eh5v4TP4-X5BtjlYaq7UKLcBsrzserE4TQN1K0P3lqHnug?e=Mn8LzWTEXT. The observed hydrological, meteorological, and salinity data used in this study were obtained from the Royal Irrigation Department of Thailand (RID, www.rid. go.th), the National Hydroinformatics Data Center of Thailand (NHC, www.thaiwater.net), the Metropolitan Waterworks Authority (MWA, www.rwc.mwa.co.th), the Department of Drainage and Sewerage of Bangkok (DDS, https://dds.bangkok.go.th), and the Thai Meteorological

Department (TMD, https://tmd.go.th).

*Author contributions.*  S.S., S.T. and S.K obtained the data, performed the analyses, developed the simulations, and evaluated the simulation results. S.S. and S.T. contributed to analysis design, concept development, and interpretation of results. All authors contributed to the writing and reviewing of the manuscript.

*Competing interests.*  The authors declare no conflict of interest.

*Acknowledgements.*  The authors gratefully acknowledge the Royal Irrigation Department of Thailand (RID), the National Hydroinformatics Data Center of Thailand (NHC), the Metropolitan Waterworks Authority (MWA), and the Department of Drainage and Sewerage of Bangkok (DDS) for providing and supporting the hydrological, meteorological, and salinity data used in this study. We also thank the Thai Meteorological Department (TMD) for providing the summary of rainfall characteristics in central Thailand.



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
