# Peer review of "Drought and Salinity Intrusion in the Lower Chao Phraya River: Variability Analysis and Modeling Mitigation Approaches"

_EGUsphere, 2024_

## Author Comment (AC2)

**This paper addresses the problem of salinity intrusion in the Lower Chao Phraya River in Thailand, by combining local observations and numerical modeling. Their study can be applied to estuarine systems worldwide and represents a complete approach for building a valuable forecasts system and for understanding the key drivers of salinity intrusions.**

**I suggest the Author to make few modifications in the following points, especially to provide more details in the methods:**

**Response:**

We sincerely thank the reviewer for the constructive and supportive feedback. We appreciate the recognition of the study's relevance and have carefully addressed all comments. The manuscript has been revised accordingly, with specific responses and corresponding changes detailed below.

**Study Area:**

**Figure 1: Please describe the figure more in detail (e.g. what is the color bar representing?), add a reference length scale bar and add a reference map to identify the position of this region, with respect to a broader region (e.g. whole Thailand and neighboring countries).**

**Response:**

We thank the reviewer for the helpful suggestion regarding Figure 1. The figure has been revised accordingly to improve clarity and geographic context (see attachment).

Specifically:

A reference length scale bar has been added.

An inset map now provides broader spatial context, showing the location of the study area within Thailand and neighboring countries.

The color bar and elevation shading are now described in the caption to clarify the topographic gradient (in meters above mean sea level).

These revisions aim to enhance the readability and usefulness of the figure for an international audience.

**Lines 66-70: Repetition of the introduction, I suggest to remove this lines.**

**Response**

The repetitive content in lines 66–70 has been removed as suggested

**Lines 81-85: Are there any references for such observational data? Please add them, whether present**

**Response**

We thank the reviewer for this suggestion. In response, we have added appropriate references and expanded the description of the hydrological background for the Lower Chao Phraya River (LCPYR) in the revised manuscript. The revised text now reads:

"The total inflow to the LCPYR is primarily regulated by two major upstream dams: the Chao Phraya (CPY) Dam and the Rama VI Dam. Observational data obtained from the National Hydroinformatics Data Center of Thailand (NHC, www.thaiwater.net; accessed on 1 April 2025) indicate that the CPY Dam releases an average dry-season inflow of approximately 80 $m^3$/s, which can peak at over 2,000 $m^3$/s during flood events. This seasonal variation reflects the broader monsoonal influence on river discharge. Similar patterns have been described in long-term records by Bidorn et al. (2021), who analyzed 70 years of hydrological data for the delta. The consistently low dry-season flows may also reflect operational practices aimed at maintaining dam stability and ensuring a minimum downstream water level, as suggested by Molle et al. (2001). Rama VI Dam contributes additional flow, with observed discharges ranging from approximately 20 $m^3$/s during dry periods to 800 $m^3$/s during high-flow conditions. Furthermore, downstream water abstraction for metropolitan water supply, particularly in the Bangkok Metropolitan Region, reduces net inflow by an average of 55 $m^3$/s (Pokavanich and Guo, 2024). Although additional lateral inflows and withdrawals exist, the lack of systematic in situ measurements introduces uncertainty into the full assessment of the river's water balance."

References added in the revised manuscript:

Bidorn, B., Sok, K., Bidorn, K., & Burnett, W.C. (2021). An analysis of the factors responsible for the shoreline retreat of the Chao Phraya Delta (Thailand). Science of the Total Environment, 769, 145253. https://doi.org/10.1016/j.scitotenv.2021.145253

Molle, F., Chompadist, C., Srijantr, T., & Keawkulaya, J. (2001). Dry-season water allocation and management in the Chao Phraya Delta. DORASDELTA Research Report, DORAS Center, Kasetsart University.

**Data and Methods:**

**Lines 112-113: How data are detrended? Please add some information on the method used to detrend time series.**

**Response**

We thank the reviewer for the comment. As requested, we have added details on the detrending method in the revised manuscript. Specifically, we applied a LOWESS (Locally Weighted Scatterplot Smoothing) filter with a smoothing parameter of f = 0.01 to extract the trend component. The time series spans from January 1, 2015, to May 14, 2021, and the smoothing parameter corresponds to a window of approximately 25 days within this period. The detrended signal was then obtained by subtracting the LOWESS-derived trend from the original time series.

**Equations 1 and 2: How the rolling mean is evaluated. Is there any variable transformation for daily streamflow and rainfall before computing the index or daily rainfall and streamflow have already a gaussian distribution? What period the rolling mean is referred to (SPI3, SPI6, …)? Please be more accurate in the description of SDI and SPI.**

**Response**

We thank the reviewer for highlighting this important point. We acknowledge that our original explanations may have caused confusion, and we have revised Sections 3.2 and 3.3 to provide clearer definitions and a more precise description of the indices used.

In the revised manuscript, we define the Rolling Standardized Discharge Anomaly (RSDA) and Rolling Standardized Precipitation Anomaly (RSPA) as follows:

$RSDA(t) = [Q(t) - Q\_mean\_dry] / \sigma\_Q\_rolling(t)$
$RSPA(t) = [P(t) - P\_mean\_dry] / \sigma\_P\_rolling(t)$

Where:

- $Q(t)$ and $P(t)$ are daily streamflow and precipitation at time t,

- $Q\_mean\_dry$ and $P\_mean\_dry$ are dry-season means,

- $\sigma\_Q\_rolling(t)$ and $\sigma\_P\_rolling(t)$ are 60-day rolling standard deviations centered on t.

This approach does not assume Gaussian-distributed input or involve prior data transformation. Instead, it captures short-term variability relative to local dry-season conditions, making it more responsive to non-stationary hydrological extremes.

Unlike traditional indices such as SDI or SPI (e.g., SPI-3, SPI-6), which rely on long-term aggregates and fitted parametric distributions, our indices use daily data and rolling statistics to support near-real-time monitoring.

Although the previous version did not clearly reflect it, the correct formulation was used consistently in our analysis. These definitions and conceptual distinctions have now been clearly incorporated into the revised manuscript to improve clarity and precision.

**Line 141: Please check the site link. In my case, it is not working.**

**Response**:

We thank the reviewer for pointing this out. The link has been corrected to the appropriate source: rwc.mwa.co.th.

**Line 225: What is the GOFS 3.1 reanalysis product (HYCOM)? Please add a reference or further explanation.**

**Response**

Thank you for the comment. We have clarified in the manuscript that the Global Ocean Forecasting System (GOFS) v3.1, produced using the HYbrid Coordinate Ocean Model

(HYCOM), provides global ocean forecast data. We used its salinity forecast product to define the downstream boundary condition. Data are available at:

https://www.hycom.org/dataserver/gofs-3pt1/analysis

https://www7320.nrlssc.navy.mil/GLBhycomcice1-12/

(accessed April 8, 2025)

**Results and discussion:**

**Generally, sections 6.3.2 and section 6.3.3 are very difficult to follow, due to many experiments and acronyms. Tables help, but the reading would be improved by summarizing such sections and maybe introducing a schematic summarizing all the experiments.**

**Response**

We thank the reviewer for the thoughtful and constructive comment regarding the complexity and readability of Sections 6.3.2 and 6.3.3. We acknowledge that the density of scenario testing, acronyms, and sensitivity analysis may have made these sections difficult to follow.

In response, we have thoroughly revised both sections to enhance clarity and coherence. Specifically:

The narrative has been streamlined by reducing repetitive technical content and grouping related findings more clearly.

The sensitivity analysis is now structured around the key components of the net inflow equation, with clear topic sentences and transitions to guide the reader.

The January 2020 salinity intrusion case study has been moved to precede the sensitivity experiments, providing real-world context and motivating the tested parameters.

Quantitative results are more clearly linked to Table 2 and supplementary figures, improving readability without overwhelming the main text.

A concluding summary has been added to Section 6.3.3 to synthesize results within the broader conceptual framework of drought-dependent and relaxation-driven salinity dynamics.

Finally, we have added a schematic diagram summarizing the experimental setup, inputs, and modeled outcomes to support comprehension, as suggested.

We believe these revisions significantly improve the readability and logical flow of the manuscript, and we sincerely appreciate the reviewer's guidance in strengthening this section.

**Figure 5: Are horizontal lines marking the periods of 14, 182, and 365 days? I am not sure about the 182 days line.**

**Response:**

Thank you for your observation. The horizontal lines marking periods of 14, 182, and 365 days have now been verified and clarified in the figure and caption. The 182-day line has been corrected to reflect the appropriate sub-annual cycle.

**Line 354: Please define Higher High Water and Lower High Water**

**Response:**

Definitions for Higher High Water (H.H.W.) and Lower High Water (L.H.W.) have been added to the manuscript as follows:

Higher High Water (H.H.W.): The higher of the high tides occurring on a given day. In semidiurnal tidal regimes, two high tides typically occur per day.

Lower High Water (L.H.W.): The lower of the high tides occurring on the same day.

**Figure 6: Please substitute cms unit with m3/s**

**Response**

The unit "cms" has been replaced with the correct "$m^3$/s" notation throughout the figure and caption.

**Line 493: correct de-pending**

**Response**

The hyphenation error "de-pending" has been corrected.

**Line 551: Non-tidal Sea Level Effects: missing bold text**

**Response**

The missing bold text for the heading "Non-tidal Sea Level Effects" has been corrected.

**Line 572: correct mitiga-tion**

**Response**

The hyphenation error in "mitiga-tion" has been corrected.

**Line 602: correct  cli-mate**

**Response**

The word "cli-mate" has been corrected to "climate".

[Figure]

**Station Definition**

| Code | Name | Hydro. | WQ. |
|------|------|--------|-----|
| S1 | South Bangkok | Water elev. | Salinity |
| S2 | Khlong Lat Pho | Water elev. | Salinity |
| S3 | Memorial Bridge | Water elev. | Salinity |
| S4 | Wat Sai Ma Nuea | Water elev. | Salinity |
| S5 | Sam Lae | Water elev. | Salinity |
| T1 | Tha Chin | Water elev. | - |
| Q1 | CPY Dam | Runoff | - |
| Q2. | Rama VI Dam | Runoff | - |
| C1. | Rama VIII Bridge | Tidal Current | - |
| C2. | Wat Khema | Tidal Current | - |
| C3. | Bang Sai | Tidal Current | - |

Lao

Thailand

Myanmar

LCPY

GOT

Cambodia

Chainat

Lower Chao Phraya River

Pasak River

Lop Buri

Q1

Q2

Ayutthaya

N

Bangkok Airport

m(aMSL)

20

15

10

0

-5

-10

Bangkok Metropolis

Tha Chin River

Mae Klong River

T1

Bang Pakong River

1:87000

0 km    20 km    40 km

---

## Author Response (AR2)

Subject: Submission of Revised Manuscript (Major Revision) to Hydrology and Earth System Sciences (HESS)

Dear Editor,

We sincerely thank you and the reviewers for your constructive feedback on our manuscript entitled "Drought and Salinity Intrusion in the Lower Chao Phraya River: Variability Analysis and Modeling Mitigation Approaches". Following the reviewer comments, we have comprehensively revised our manuscript and are pleased to submit this updated version for consideration of publication in Hydrology and Earth System Sciences (HESS).

To clearly demonstrate our revisions, we have provided a tracked-changes version of the manuscript. Specifically:

- Deleted text is shown in red with strikethrough.
- Inserted text is highlighted in blue.
- Structural and editorial changes have been extensively applied throughout the manuscript to enhance readability, accuracy, and overall scientific clarity.

Major changes include improved definitions and justifications of the rolling-window standardized indices (RSDA and RSPA), additional clarifications and corrections based on reviewers' requests, and streamlining of the results and discussion sections. Furthermore, schematic diagrams have been included in supplementary materials (Figures S9 and S10) to summarize experimental designs and outcomes, enhancing the manuscript's clarity and reader-friendliness.

The subsequent pages of this document provide detailed point-by-point responses addressing each reviewer's comments, clearly indicating how and where the suggestions have been incorporated.

We appreciate the thoughtful feedback provided, which has significantly improved the quality and clarity of our manuscript. We look forward to your positive consideration.

Sincerely,

Sirod Sirisup

Data-driven Simulation and Systems Research Team

National Electronics and Computer Technology Center (NECTEC)

112 Phahonyothin Road, Khlong Nueng, Khlong Luang District

Pathumthani 12120, Thailand

Corresponding Author: Sirod Sirisup (sirod.sirisup@nectec.or.th)

**Saltwater intrusion is a serious concern for coastal regions. In this paper, the attention is paid to the Lower Chao Phraya River. The effects of hydrological droughts on saltwater intrusion is investigated by both wavelet analysis and hydrodynamic models. In general, the paper presents an interesting case study. There are five comments for further improvements of the paper:**

We sincerely thank the reviewer for the careful review, insightful comments, and valuable suggestions, which have greatly improved the clarity and quality of our manuscript. We are encouraged by the reviewer's positive recognition of the importance and relevance of our study focusing on saltwater intrusion in the Lower Chao Phraya River.

Below, we provide a detailed point-by-point response to each of the reviewer's comments and clearly outline the revisions made in the manuscript to address them.

**-First of all, saltwater intrusion is in general an old topic. There are quite a number of previous publications. For a few examples, please refer to Liu et al. (2020), Liu et al. (2019) and Weng et al. (2024). The authors may want to highlight what new insights this paper presents.**

**Response:**

We thank the reviewer for highlighting important previous studies and encouraging us to clarify the novel contributions of our work. In response, we have revised the manuscript to explicitly emphasize the unique insights provided by this study. Specifically, our work advances understanding of saltwater intrusion mechanisms in the Lower Chao Phraya River (LCPYR) through the following key findings (in Conclusions section, Lines 692-707 and Lines 721-728):

**Characterization of drought-dependent and relaxation relationships**

We elucidate how basin-scale drought magnitudes influence salinity intrusion (drought-dependent mechanism) and highlight the complexities arising when local hydro-climatic conditions temporarily interrupt this linkage (relaxation mechanism). The drought-dependent mechanism underscores the importance of basin-scale hydro-climatic variability, while the relaxation relationship emphasizes the significance of local climate variability.

**Pinpointing salinity intrusion sources using a physically based numerical model**

The numerical model explicitly identifies and prioritizes the key drivers of saltwater intrusion. It also clarifies the relaxation mode suggested by wavelet analysis, highlighting previously underappreciated factors such as urban runoff and freshwater supply management near the river's downstream end.

**Insight into Improving the Effectiveness of Freshwater Redistribution (Water Hammer Technique)**

We demonstrate and reevaluate the effectiveness of strategically timed freshwater redistribution ("water hammer"), emphasizing duration and discharge volume. This technique, previously unexplored in the LCPYR context, is critically assessed, and we provide revised recommendations alongside proactive and long-term strategies for managing salinity intrusion.

These additions clarify our study's contribution, differentiating it from existing literature and providing practical insights into improved management practices.

**-The second comment relates to the first one. Every case study is unique to some extent. Therefore, the authors may want to illustrate some unique characteristics of the Lower Chao Phraya River. In particular, the "study area" and "data" can be presented in the same section. In this way, people can better understand the background.**

**Response:** We thank the reviewer for highlighting the importance of clearly illustrating the unique characteristics of the Lower Chao Phraya River (LCPYR). To address this suggestion, we have revised the manuscript to explicitly emphasize these unique aspects (Lines 82-92) , as follows.

*" Topographically, prominent estuaries of the river deltas in Indochina and East Asia, such as the Irrawaddy, Mekong, and Pearl Rivers, are positioned on the windward side relative to the prevailing monsoon winds. Consequently, their seasonal flow regimes are influenced by both monsoon and synoptic weather systems (Fan and Luo, 2019; Chen et al., 2024; Besset et al., 2017; Sirisena et al., 2021; Weng et al., 2024, 2020). In contrast, the Chao Phraya River and its estuary lie on the leeward side of the mountain ranges (e.g. Thongchai-Tenasserin and Annamite ranges) and plateau, causing the rainfall-runoff regime to be predominantly controlled by synoptic weather systems, i.e., tropical cyclones and monsoon troughs rather than direct monsoon winds (Tsai et al., 2015; Tomkratoke and Sirisup, 2022). While the CPYB dry season generally coincides with that of the broader mainland region, a unique local hydro-climatic feature is the occasional occurrence of wet episodes triggered by southerly or southeasterly winds across the inner GOT during the dry season. Although the underlying mechanisms driving this phenomenon remain understudied, it notably induces temporary wet periods that significantly accelerate the dilution of salinity levels in the LCPYR during otherwise dry conditions."*

**In particular, the "study area" and "data" can be presented in the same section. In this way, people can better understand the background.**

**Response**: Done, we have integrated the "Study Area" and "Data" sections into one comprehensive section.

**-Thirdly, the methods of wavelet analysis and hydrodynamic models are currently presented in two sections. The authors may want to combine them into one section.**

**Response**: Done, we have combined the previously separate sections detailing wavelet analysis and hydrodynamic modeling into a unified Methodology section.

**-The fourth comment relates to the third one. What new findings are made through the combined use of wavelet analysis and hydrodynamic models? In particular, can some early warnings be developed from the combined use?**

**Response:** We thank the reviewer for this insightful comment to clarify new insights arising from the combined application of wavelet analysis and hydrodynamic modeling. We have revised the manuscript to clearly highlight these findings (in Conclusions section, Lines 708-720), which we summarize below:

**New Insight from Combined Methods:**

A key finding from the combined wavelet and hydrodynamic analyses is that local hydro-climatic variability, especially urban runoff from nearby watersheds, can significantly enhance salinity dilution even during severe drought conditions. This local influence, which has not been previously reported in the context of the Lower Chao Phraya River, corresponds directly with the relaxation relationship identified through wavelet analysis. Our numerical simulations further validated the critical importance of this urban runoff influence. Neglecting this factor resulted in reduced model accuracy, underscoring the potential for misinformed mitigation strategies and inefficient utilization of upstream freshwater resources.

**Implications for Early Warning Systems:**

Wavelet analysis effectively reveals the historical evolution of variability modes, including active and inactive phases of drought. However, its predictive capability is inherently limited by the wavelet's cone of influence (COI), restricting its practical application for forecasting and early warning. An alternative approach proposed in our study involves directly monitoring drought characteristics such as duration and magnitude through standardized indices (e.g., the rolling standardized discharge anomaly (RSDA) and rolling standardized precipitation anomaly (RSPA)). Coupled with insights from numerical modeling or predictive systems, these indices may help anticipate future salinity conditions, offering practical potential for early warning applications. Nonetheless, further investigation is needed to refine and validate this integrated forecasting approach.

**-Fifthly, the abstract tells that "… offer essential insights to guide management strategies and the development of prediction tools for the LCPYR and surrounding regions." More details are in demand.**

**Response:** We appreciate the reviewer's valuable suggestion to provide additional details regarding the management strategies and prediction tools outlined in our abstract.

To address this comment, we have revised the manuscript to clearly specify and elaborate on the practical implications of our findings (Lines 645-678). In particular, we now detail:

- **Immediate mitigation measures:** We illustrate the application and refinement of the freshwater redistribution technique, termed the "water hammer," focusing on optimizing the duration and volume of freshwater releases to effectively mitigate saltwater intrusion during emergent drought periods.

- **Long-term proactive strategies:** We highlight strategies incorporating the influence of urban runoff and tributary inflows on salinity intrusion. Furthermore, we discuss the potential development of a robust predictive framework that integrates hydrological and climatic variability to enhance long-term planning and early warning capabilities.

These elaborations are explicitly provided in the revised manuscript, offering a clearer understanding of the practical significance and applicability of our research findings.

**References:**

Liu, D., Chen, X. and Lou, Z., 2010. A model for the optimal allocation of water resources in a saltwater intrusion area: a case study in Pearl River Delta in China. Water resources management, 24, pp.63-81.

Liu, B., Peng, S., Liao, Y. and Wang, H., 2019. The characteristics and causes of increasingly severe saltwater intrusion in Pearl River Estuary. Estuarine, coastal and shelf science, 220, pp.54-63.

Weng, P., Tian, Y., Zhou, H., Zheng, Y. and Jiang, Y., 2024. Saltwater intrusion early warning in Pearl River delta based on the temporal clustering method. Journal of Environmental Management, 349, p.119443.

**Response:**

We appreciate the reviewer's suggestion. The recommended references have been included (Lines 67 and 85) to acknowledge key contributions from related studies in other estuarine systems.

**This paper addresses the problem of salinity intrusion in the Lower Chao Phraya River in Thailand, by combining local observations and numerical modeling. Their study can be applied to estuarine systems worldwide and represents a complete approach for building a valuable forecasts system and for understanding the key drivers of salinity intrusions.**

**I suggest the Author to make few modifications in the following points, especially to provide more details in the methods:**

**Response:**

We sincerely thank the reviewer for the constructive and supportive feedback. We appreciate the recognition of the study's relevance and have carefully addressed all comments. The manuscript has been revised accordingly, with specific responses and corresponding changes detailed below.

**Study Area:**

**Figure 1: Please describe the figure more in detail (e.g. what is the color bar representing?), add a reference length scale bar and add a reference map to identify the position of this region, with respect to a broader region (e.g. whole Thailand and neighboring countries).**

**Response:**

We thank the reviewer for the helpful suggestion regarding Figure 1. The figure has been revised accordingly to improve clarity and geographic context (see attachment).

Specifically:

A reference length scale bar has been added.

An inset map now provides broader spatial context, showing the location of the study area within Thailand and neighboring countries.

The color bar and elevation shading are now described in the caption to clarify the topographic gradient (in meters above mean sea level).

These revisions aim to enhance the readability and usefulness of the figure for an international audience.

**Lines 66-70: Repetition of the introduction, I suggest to remove this lines.**

**Response**

The repetitive content in lines 66–70 has been removed as suggested

**Lines 81-85: Are there any references for such observational data? Please add them, whether present**

**Response**

We thank the reviewer for this suggestion. In response, we have added appropriate references and expanded the description of the hydrological background for the Lower Chao Phraya River (LCPYR) in the revised manuscript (Lines 105-117). The revised text now reads:

"The total inflow to the LCPYR is primarily regulated by two major upstream dams: the Chao Phraya (CPY) Dam and the Rama VI Dam. Observational data obtained from the National Hydroinformatics Data Center of Thailand (NHC, https://www.thaiwater.net, accessed on 1 April 2025) indicate that the CPY Dam releases an average dry-season inflow of approximately 80 $m^3$/s, which can peak at over 2,000 $m^3$/s during flood events. This seasonal variation reflects the broader monsoonal influence on river discharge. Similar patterns have been reported in long-term hydrological records, such as the 70-year analysis by (Bidorn et al., 2021), which highlights seasonal discharge trends and their implications for delta stability. The persistently low flows observed during dry seasons may also result from dam operation strategies intended to preserve structural integrity and ensure downstream water availability, as suggested by (Molle et al., 2001). The Rama VI Dam contributes additional discharge, ranging from approximately 20 $m^3$/s in dry conditions to as much as 800 $m^3$/s during high-flow periods. Furthermore, downstream water abstraction, especially for municipal water supply in the Bangkok Metropolitan Region, reduces net inflow by an average of 55 $m^3$/s (Pokavanich and Guo, 2024). Although additional lateral inflows and withdrawals exist, the lack of systematic in-situ measurements introduces uncertainty in fully quantifying the river's water balance."

References added in the revised manuscript:

Bidorn, B., Sok, K., Bidorn, K., & Burnett, W.C. (2021). An analysis of the factors responsible for the shoreline retreat of the Chao Phraya Delta (Thailand). Science of the Total Environment, 769, 145253. https://doi.org/10.1016/j.scitotenv.2021.145253

Molle, F., Chompadist, C., Srijantr, T., & Keawkulaya, J. (2001). Dry-season water allocation and management in the Chao Phraya Delta. DORASDELTA Research Report, DORAS Center, Kasetsart University.

**Data and Methods:**

**Lines 112-113: How data are detrended? Please add some information on the method used to detrend time series.**

**Response**

We thank the reviewer for the comment. As requested, we have added details on the detrending method in the revised manuscript (Lines 146-149). Specifically, we applied a Locally Weighted Scatterplot Smoothing (LOWESS) filter with a smoothing parameter of f = 0.01 to extract the trend component. The time series spans from January 1, 2015, to May 14, 2021, and the smoothing parameter corresponds to a window of approximately 25 days within this period. The detrended signal was then obtained by subtracting the LOWESS-derived trend from the original time series.

**Equations 1 and 2: How the rolling mean is evaluated. Is there any variable transformation for daily streamflow and rainfall before computing the index or daily rainfall and streamflow have already a gaussian distribution? What period the rolling mean is referred to (SPI3, SPI6, …)? Please be more accurate in the description of SDI and SPI.**

**Response**

We thank the reviewer for highlighting this important point. We acknowledge that our original explanations may have caused confusion. We have revised Sections 2.2.2 and 2.2.3 (Lines 150-192) to align the manuscript text with the indices as actually implemented.

In the revised manuscript, we define the Rolling Standardized Discharge Anomaly (RSDA) and Rolling Standardized Precipitation Anomaly (RSPA) as follows:

$$RSDA(t) = [Q(t) - Q\_mean\_dry] / \sigma\_Q\_rolling(t)$$
$$RSPA(t) = [P(t) - P\_mean\_dry] / \sigma\_P\_rolling(t)$$

Where:

- $Q(t)$ and $P(t)$ are raw daily streamflow and precipitation at time $t$,

- Q_mean_dry and P_mean_dry are the climatological dry-season means (November–June),

- $\sigma\_Q\_rolling(t)$ and $\sigma\_P\_rolling(t)$ are 60-day rolling standard deviations centered on $t$.

Both RSDA and RSPA are defined directly from daily data, without any prior transformation or parametric fitting, and use a fixed dry-season mean with a rolling variance. Only the variance is updated locally; the mean remains fixed to highlight anomalies during the months most critical for saltwater intrusion. The 60 day window was chosen to match key sub-seasonal drivers (spring–neap tidal cycles and the Madden–Julian Oscillation) rather than to emulate SPI-3 or SPI-6 aggregation periods. These formulations do not presuppose Gaussian inputs, no data transformation or distributional fit is performed, yet they produce robust, context-sensitive indicators of hydrological and meteorological anomaly under non-stationary conditions. Unlike traditional SPI or SDI, which aggregate over multi-month totals and fit probability distributions, RSDA and RSPA deliver daily, near-real-time anomalies responsive to evolving drought and salinity-intrusion threats.

Although the previous version did not clearly reflect it, the correct formulation was used consistently in our analysis. These definitions and conceptual distinctions have now been clearly incorporated into the revised manuscript to improve clarity and precision.

**Line 141: Please check the site link. In my case, it is not working.**

**Response**:

We thank the reviewer for pointing this out. The link has been corrected to the appropriate source: rwc.mwa.co.th.

**Line 225: What is the GOFS 3.1 reanalysis product (HYCOM)? Please add a reference or further explanation.**

**Response**

Thank you for the comment. We have clarified in the manuscript (Lines 281-285) that the Global Ocean Forecasting System (GOFS) v3.1, produced using the HYbrid Coordinate Ocean Model (HYCOM), provides global ocean forecast data. We used its salinity forecast product to define the downstream boundary condition. Data are available at:

https://www.hycom.org/dataserver/gofs-3pt1/analysis

https://www7320.nrlssc.navy.mil/GLBhycomcice1-12/

(accessed April 8, 2025)

**Results and discussion:**

**Generally, sections 6.3.2 and section 6.3.3 are very difficult to follow, due to many experiments and acronyms. Tables help, but the reading would be improved by summarizing such sections and maybe introducing a schematic summarizing all the experiments.**

**Response**

We thank the reviewer for the thoughtful and constructive comment regarding the complexity and readability of Sections 6.3.2 and 6.3.3. We acknowledge that the density of scenario testing, acronyms, and sensitivity analysis may have made these sections difficult to follow.

These sections have been renumbered to Sections 4.3.2 and 5 in the revised manuscript. We have:

- Streamlined the narrative by reducing repetitive technical content and grouping related findings more clearly.
- Structured the sensitivity analysis around the key components of the net-inflow equation, with clear topic sentences and transitions to guide the reader.
- Moved the January 2020 salinity intrusion case study to precede the sensitivity experiments, providing real-world context and motivation.
- Linked quantitative results to Table 2 and Figures S9 and S10 to improve readability without overloading the main text.
- Added a concluding summary to each section to synthesize results within the broader framework of drought-dependent and relaxation-driven salinity dynamics.
- Included a schematic diagram (Figures S9 and S10) summarizing the experimental setup, inputs, and modeled outcomes to support comprehension.

We believe these revisions significantly improve the readability and logical flow of the manuscript, and we sincerely appreciate the reviewer's guidance in strengthening this section.

**Figure 5: Are horizontal lines marking the periods of 14, 182, and 365 days? I am not sure about the 182 days line.**

**Response:**

Thank you for your observation. The horizontal lines marking periods of 14, 182, and 365 days have now been verified and clarified in the figure and caption. The 182-day line has been corrected to reflect the appropriate sub-annual cycle.

**Line 354: Please define Higher High Water and Lower High Water**

**Response:**

Definitions for Higher High Water (HHW) and Lower High Water (LHW) have been added to the manuscript as follows (Lines 415-418):

These terms are defined as follows: HHW refers to the higher of the two daily high tides, while LHW denotes the lower of the two. This distinction is particularly relevant in semidiurnal tidal regimes, which feature two high and two low tides each day (National Oceanic and Atmospheric Administration (NOAA), 2020).

Reference:

National Oceanic and Atmospheric Administration (NOAA): Tidal Datums and Their

Applications, Tech. rep., Center for Operational Oceanographic Products and Services (CO-

OPS), 2020,

https://tidesandcurrents.noaa.gov/publications/tidal_datums_and_their_applications.

pdf, accessed April 8, 2025.

**Figure 6: Please substitute cms unit with m3/s**

**Response**

The unit "cms" has been replaced with the correct "m$^3$/s" notation throughout the figure and caption.

**Line 551: Non-tidal Sea Level Effects: missing bold text**

**Response**

The missing bold text for the heading "Non-tidal Sea Level Effects" has been corrected.

**Line 493: correct de-pending**

**Line 572: correct mitiga-tion**

**Line 602: correct  cli-mate**

**Response**

We've corrected "de-pending," "mitiga-tion," and "cli-mate" to "depending," "mitigation," and "climate," and have fixed all other improper hyphenations throughout the manuscript